# The role of geological mouth islands on the morphodynamics of back-barrier tidal basins

Yizhang Wei[1], Yining Chen[2], Jufei Qiu[3], Zeng Zhou[1,4], Peng Yao[1], Qin Jiang[4], Zheng Gong[1], Giovanni Coco[5,1], Ian Townend[1,6] and Changkuan Zhang[1]

[1]State Key Laboratory of Hydrology-Water Resources and Hydraulic Engineering, Hohai University, Nanjing 210098, China

[2]Second Institute of Oceanography, Ministry of Natural Resources, Hangzhou 310012, China

[3]East Sea Marine Environmental Investigating and Surveying Centre, State Oceanic Administration of China, Shanghai 310115, China

[4]Jiangsu Key Laboratory of Coast Ocean Resources Development and Environment Security, Hohai University, Nanjing 210098, China

[5] Faculty of Science, University of Auckland, Private Bag 92019, Auckland, New Zealand

[6] Ocean and Earth Sciences, University of Southampton, Southampton SO17 1BJ, UK

*Correspondence to*: Dr Zeng Zhou (zeng.zhou@hhu.edu.cn)

**Abstract.** The morphodyamics of back-barrier tidal basins have been extensively investigated by numerical modelling, but the influence of mouth islands (which may be submerged under future sea level rise) has been rarely explored. Using the Dongshan Bay in southern China as a reference site, we numerically explore the effects of geological constraints (i.e., islands) on the long-term morphodynamics of back-barrier basins. Model results indicate that the spatial configuration of mouth islands can considerably affect the morphological development of tidal basins. The presence of mouth islands can increase both the current velocity and the residual current by narrowing the inlet cross-sectional area, resulting in more sediment suspension and transport. Meanwhile, erosion tends to occur in the tidal basin and sedimentation occurs in the ebb-delta area, and the erosion (or sedimentation) volume is larger with the presence of more mouth islands. Further, the spatial distribution of mouth islands can also considerably affect tidal basin evolution: the basin-side mouth islands tend to cause more basin erosion with higher tidal currents and more sediment transport, while the delta-side ones may play a hindering role resulting in sediment deposition in the basin. Finally, larger tidal prisms are observed in basins with more mouth islands and those with basin-side mouth islands, suggesting that the number and location of mouth islands can also affect the relation between tidal prism and inlet cross-sectional area. This modelling study furthers the understanding of barrier basin morphodynamics affected by mouth islands and informs management strategies under a changing environment.

## 1 Introduction

During the post glacial sea-level rise, a large number of low-lying basins and valleys were submerged, forming various back-barrier systems accounting for 10-15% of the world's coastline (De Swart and Zimmerman, 2009; Fitzgerald and Miner, 2013). Back-barrier systems are easily found around the world, such as the Wadden Sea (Wang et al., 2012), the Venice Lagoon (Feola et al., 2005), and the Massachusetts Bay (Rosen and Leach, 1987). Knowledge on the morphodynamic processes and the evolution of barrier systems is of great significance for better protection and management of this type of coastal zones.

Morphodynamics of back-barrier systems are affected by the interaction of various factors, including hydrodynamic processes (e.g., tides and waves), biological activities (e.g., presence of mangroves and/or salt marshes), climate change

(e.g., global warming and sea level rise) and anthropogenic activities, such as land reclamation and artificial construction (Murray et al., 2008). These processes result in an ever-changing morphology of tidal channels, tidal flats and flood/ebb deltas (Wang et al., 2012). Coastal morphodynamics are typically characterized by the two-way feedback between hydrodynamics and topography. Hydrodynamics can shape the geomorphic characteristics of coastal landforms through sediment transport, while the changed landforms can also feedback to hydrodynamics, forming a morphodynamic loop which eventually drives the system to some sort of dynamic equilibrium state (Coco et al., 2013; Zhou et al., 2017).

In the last decades, many studies have been carried out on the morphodynamics of tidal barrier systems. Otvos (1981) analysed the rock drilling data of the Mississippi barrier island chain, and reported that sand bar drifting down from the core area of the island may be necessary for the early stage of barrier island survival. Using laboratory experiments, Stefanon et al. (2010) showed that the experimental tidal channels and tidal flats generated were comparable to that in natural back-barrier systems and sea level variations can leave morphological signatures in these systems in terms of channel network incision and retreat. Numerical modelling is another effective and virtual tool, Zhou et al. (2014b) compared the laboratory experiment and numerical simulation of the morphological evolution in barrier basins, and their results suggested that the initial bathymetry and geometric characteristics of barrier basins have a great influence on the development of tidal channels. Marciano (2005) numerically simulated the branching channel patterns observed in the Wadden Sea basins and model results indicated that tidal channel patterns were governed by the morphological characteristics (e.g., the bottom slope and the water depth) and the Shields parameters (e.g., flow strength and sediment properties). Using a similar type of model, Dastgheib et al. (2008) explored channel network formation in a multi-inlet tidal system, and model results qualitatively followed the empirical equilibrium equations, indicating that initial bathymetry and the effect of adjacent basins can significantly affect the evolution of barrier basins. Van Maanen et al. (2013a) developed a new 2D morphodynamic model and explored the effects of both tidal range and initial bathymetry on producing different morphological patterns. Several studies also highlighted the importance of wave action on the morphodynamics of back-barrier systems. For instance, Herrling and Winter (2014) simulated the sediment dynamics in the mixed energy tidal inlet systems and demonstrated that the pathway and sediment distribution are much different under fair weather and storm conditions, indicating that waves have a great influence on sediment transport pattern and morphological evolution of the back-barrier systems (see also Nahon et al. (2012)). With respect to the effect of sea level rise on back-barrier basins, Dissanayake et al. (2012) and Van Maanen et al. (2013b) designed schematized models to explore their long-term evolution, and model results suggested that sea level rise can lead to the change of sediment transport from seaward to landward and the intertidal area can reduce considerably. For river-influenced barrier systems, Zhou et al. (2014a) applied an idealized model to simulate the effects of different landscape conditions (e.g., basin shape and river inflow location) on barrier basins, suggesting that the presence of a river was fundamental for the sediment budget and the morphological evolution.

The above modelling studies have significantly advanced our understanding of the influence of various factors on back-barrier tidal basins, but few studies directly consider the role of isolated islands which are very common landforms at the inlet mouth. It remains unclear how back-barrier basins evolve with and without mouth islands. For example, Figure 1 shows two example sets of neighbouring barrier basins around the world: the Massachusetts Bay and the Plymouth Bay along the eastern coast of the USA, and the Zhaoan Bay and Jiuzhen Bay along the southern coast of China. The tidal basins in these two sets of examples have relatively close spatial distance (about 30-40km) with similar geomorphic shapes but they have developed quite different morphological patterns due to the different geological constraints at the inlet mouth. As showed in Figure 1, shallow areas with dendritic channel networks have been developed in the Plymouth Bay and the

Jiuzhen Bay. In contrast, a few islands are observed in the basins of Massachusetts Bay and the Zhaoan Bay, where tidal flats and tidal channels only developed very limitedly. Although all of these four barrier systems are semi-closed tidal basins, the existence of mouth islands may be one of the main contributors to their different morphologies. In the past few decades, a large number of studies have been carried out on the formation and classification of islands in geological research, which generally believed that islands can be classified into three types: continental islands, oceanic islands (including volcanic islands and coral islands) and alluvial islands (Liu and Liu, 2008; Huang and Zhang, 2006). The mouth islands in Figure 1 are continental islands, which are probably formed by the migration of continental plates and sea level rise in post glacial period (Fitzgerald, 1993; Jagoutz and Behn, 2013).

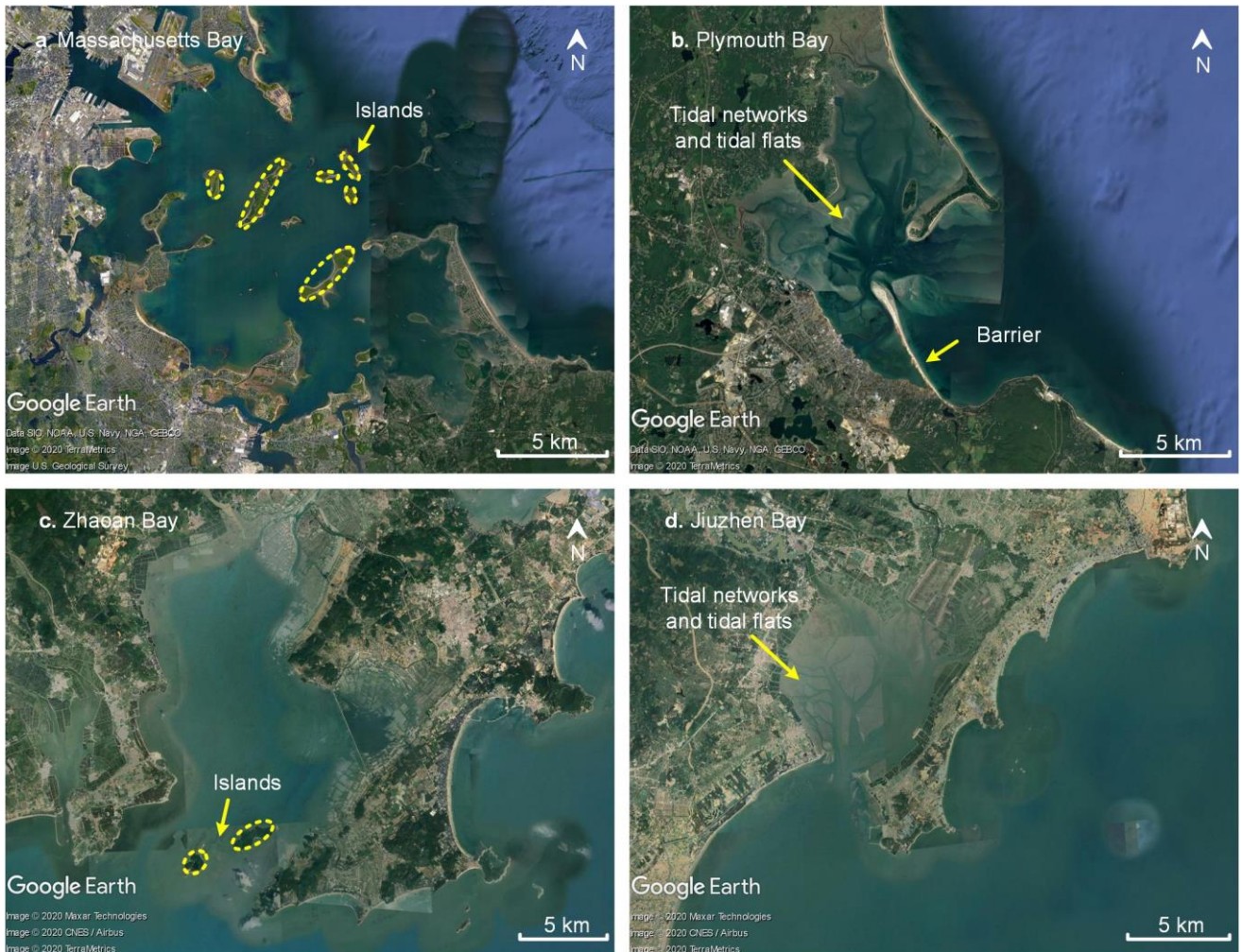

**Figure 1: Aerial view of two sets of tidal barrier basins (about 30-40 km from each other) along the eastern coast of the USA and the southern coast of China. (a) Massachusetts Bay (42° 18' 36" N, 70° 58' 12" W); (b) Plymouth Bay (42° 00' 18" N, 70° 39' 18" W); (c) Zhaoan Bay (23° 40' 12" N, 117° 18' 36" E); (d) Jiuzhen Bay (23° 59' 6" N, 117° 42' 36" E). Image © Google Earth 2020, TerraMetrics.**

This study aims to gain insight into the presence of geological mouth islands that lead to the observed different morphologies in back-barrier tidal basins. Specific research questions include: (1) What is the morphodynamic behaviour of tidal basins with varying number of mouth islands? (2) What is the role of different island locations on basin morphological evolution? To answer these questions, an idealized morphodynamic model is established, with the Dongshan Bay, China as a reference basin size, to investigate the impacts of mouth islands on the evolution of a semi-enclosed basin.

The outcome of this study can assist coastal managers and policymakers to make more sustainable management strategies for the reclamation and artificial-island construction.

## 2 Methodology

### 2.1 Model description

Based on the Delft3D open-source software, a 2D morphodynamic model is set up, which solves the coupled equations
governing tidal flow, sediment transport and bed level updating (Lesser et al., 2004; Marciano, 2005; Van Der Wegen and Roelvink, 2008). The alternative direction implicit (ADI) method is used to solve the shallow water equations for a detailed description of flow field, which is used in the calculation of sediment transport. Then the morphological change caused by the sediment transport is also fed back to the hydrodynamics at each time step. In this study, a widely adopted formula (Engelund and Hansen, 1967) is considered to calculate sediment transport. The formula of Engelund and Hansen (1967)
as follows:

$$S = \frac{0.05U^5}{g^{1/2}C^3\Delta^2 D_{50}} \tag{1}$$

where, $S$ is the total sediment transport (m$^2$/s), $U$ is the depth-averaged flow velocity (m/s), $g$ is the gravity constant (m/s$^2$), $C$ is the Chézy friction coefficient (m$^{1/2}$ s$^{-1}$), $\Delta$ is the relative density, $\Delta = (\rho_s - \rho_w)/\rho_w$, and $D_{50}$ is the median grain size of sediment (m).

In order to speed up morphodynamic calculations, a "morphological factor" (MF) is applied to following Roelvink (2006). In this approach, the sediment erosion and deposition fluxes are scaled up by a constant factor (MF) at each hydrodynamic time step to mimic morphological changes over longer duration. This approach has been extensively used in previous studies, including schematic cases (Roelvink, 2006; Van Der Wegen and Roelvink, 2008) and real-world situations (Dastgheib et al., 2008; Van Der Wegen and Roelvink, 2012). It has been suggested that the value of MF should be
determined via sensitivity experiments (Zhou et al., 2014a; Van Der Wegen and Roelvink, 2012). Some sensitivity tests with varying MF values are performed in order to select the MF value. Specifically, it is necessary to ensure that the increased bed elevation in each time-step is small enough relative to the water depth, so that the hydrodynamic process in the next time step is not significantly different from the morphological factor of application 1 (Ranasinghe et al., 2011). In this way, on the basis of ensuring the calculation accuracy, the MF value is selected as 50 to reduce the computational cost.

### 2.2 Model configurations

In this numerical experiment, an idealized model is set up with a comparable dimension as the Dongshan Bay, China, a typical barrier basin with a number of mouth islands near the inlet (Figure 2). The schematic model consists of a "stomach" shaped basin and a semi-circular open sea area with a radius of approximately 30 km (Figure 2a). For the initial basin bathymetry, an idealized central channel is set up and defined as 5 m at the landward head and linearly varying to 10 m
near the inlet. The bottom of the outer sea is linearly sloped with an elevation of 10 m near the inlet to 40 m at the sea boundary, mimicking a shallow continental shelf. The idealized bathymetry adopted in this study is intended to investigate the effect of mouth islands on the long-term morphodynamic processes and the formation of tidal channels from a flat topography. Since this study mainly focuses on the morphology inside the basin, the spatial grid has smaller grid sizes (cell size of 100 m) in the basin and larger grid sizes (cell size of 400 m) in the open sea area.

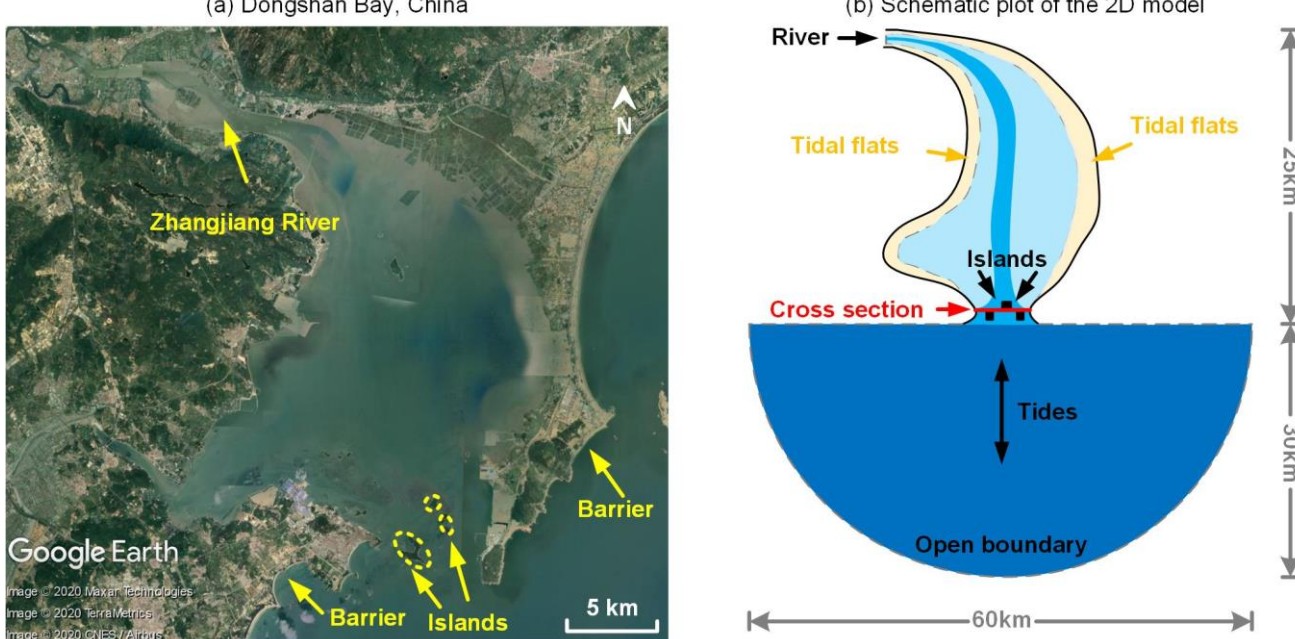

**Figure 2: (a) The reference barrier basin with three mouth islands near the inlet, Dongshan Bay (23° 49' 48" N, 117° 31' 18" E; Image © Google Earth 2020, Maxar Technologies); (b) The schematized model domain used in this study.**

The hydrodynamic processes considered in this model include tides and riverine inflow because the emphasis is on the inner basin morphology, while waves inside the basin are limited due to the sheltering of islands and hence neglected for simplicity. The river inflow is added at the landward end of tidal basin with a constant discharge of 50 m³/s. A semidiurnal harmonic tide with a tidal range of 2.4 m is specified at the southern semi-circular sea boundary following Chen et al. (1993). The sediment fraction considered in this study is non-cohesive sand only, which is the most abundant component in the Dongshan Bay and defined by a mean grain size of 135 μm Chen et al. (1993). A sediment transport boundary condition of equilibrium sediment concentration is adopted at both the sea boundary and the river boundary. This means the sediment input through the inflow boundaries can be immediately adapted to the local flow condition, ensuring the bed level near the model boundaries is almost unchanged.

Sensitivity tests have been carried out to determine some other model parameters, such as the Cheźy friction coefficient (65 m$^{1/2}$/s$^{-1}$), horizontal eddy viscosity (1 m²/s) and hydrodynamic time step (60 s).

### 2.3 Sensitivity scenarios

Islands can potentially be submerged or even disappeared due to projected sea-level rise and human activities (Webb and Kench, 2010), thus the number and location of mouth islands can be changed accordingly. However, there is still a lack of systematic understanding of the effect of varying numbers and locations of mouth islands on basin morphodynamics. Two scenarios of simulations are set up to explore their effects on basin morphologies and model configurations are shown in Figure 3. Four cases are designed to investigate the effects of island numbers (0-3) which are hereafter indicated as ''0i'', ''1i'', ''2i'' and ''3i'' for simplicity (Figure 3a-d). The other three cases are designed to explore the role of island locations which are hereafter indicated as ''IL'', ''BS'' and ''DS'' for simplicity (Figure 3e-g). In all cases, the same initial bathymetry is adopted so that model results can be compared. Besides, the shape and size of the island may also be two of the important factors affecting the morphological evolution, which are going to be investigated in future studies. For this

study, a non-erodible island with the shape of rectangle (1 km long and 1 km wide) is designed, which is assumed to mimic a rocky island near the inlet.

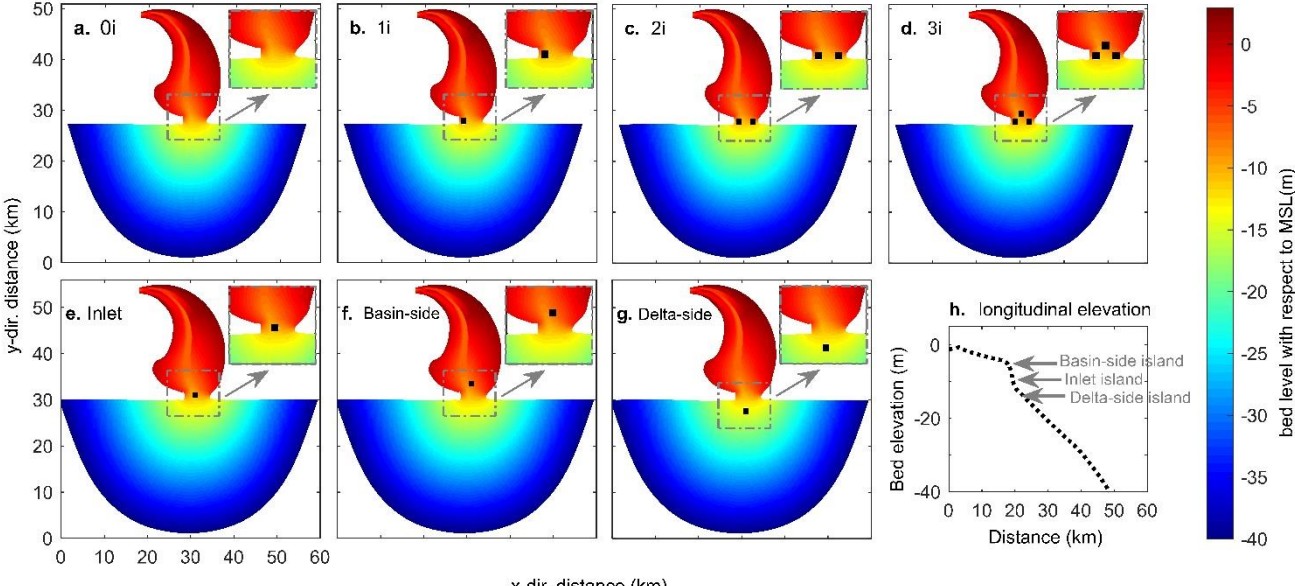

**Figure 3: Initial mode domain shape and bathymetry: (a) basin without island ("0i"); (b) basin with one island ("1i"); (c) basin with two islands ("2i"); (d) basin with three islands ("3i"); (e) the mouth island is at the inlet ("IL"); (f) the mouth island is at the basin-side near the inlet ("BS"); (g) the mouth island is at the delta-side near the inlet ("DS"); (d) longitudinal profile of the initial bed elevation and the location of mouth islands. In Figures 3a-3g, the black solid area in each plot represent the mouth islands and we zoom in part of the model domain where there are different island settings. Locations of the mouth islands are indicated by the arrows in the longitudinal profile.**

## 3 Model results

### 3.1 Influence of the number of mouth islands

Morphological evolution firstly occurs in the mouth zone where tidal currents are strongest and river input zone where there are river inflows. The initial morphodynamic development is characterized by large bathymetric changes and rapid development of tidal channels (Figure 5). In the subsequent morphological evolution, the tidal channels keep on dissecting the shallow basin through headward growth (D'alpaos, 2005). This process of branching and elongating of the channels ultimately leads to the formation of a dendritic channel network and a great many scattered sand bars. Finally, the tidal basin gradually becomes stable in shape and only minor bathymetric changes occur (Figure 4).

The presence of different numbers of mouth islands causes local differences in morphodynamic patterns near the inlet. For the case of zero island (0i), extensive erosion rapidly occurs near the inlet mouth because of strong tidal currents therein and a small-scale channel network is formed in the first 100 years (Figure 4d). After 300 years of development, the tidal network has been further developed and a large amount of sediment has been transported to the open sea, forming a complex channel network in tidal basin (Figure 4h). With the continuous morphodynamic evolution, few differences are observed in the horizontal distribution of the tidal channels, which are only gradually deepened in the vertical direction, indicating that the tidal basin has reached a stable state (Figure 4l). When a mouth island is added at the left side of the inlet, a large area of back-barrier deposition is observed behind the island, and the water depth beside the island is relatively larger. In addition, the tidal network on the left side is developed into higher intertidal area compared with the scenario of zero island (Figures 4e, 4i, 4m). If another mouth island is added at the right side of the inlet, similarly, another back-barrier deposition

is observed behind the island. The sediment in the basin is transported to near the tidal inlet owing to the larger tidal currents produced by the narrower tidal inlet (Figures 4f, 4j, 4n). Further increasing the number of mouth islands, larger spatial scale tidal network is observed in the first 100 years. And more erosion occurs in the tidal inlet mouth, indicating the hydrodynamic conditions are even stronger (Figures 4g, 4k, 4o). While in the upstream zone, small differences are observed in four cases, indicating that the effect of hydrodynamic on this area is limited (Figures 4p-4s).

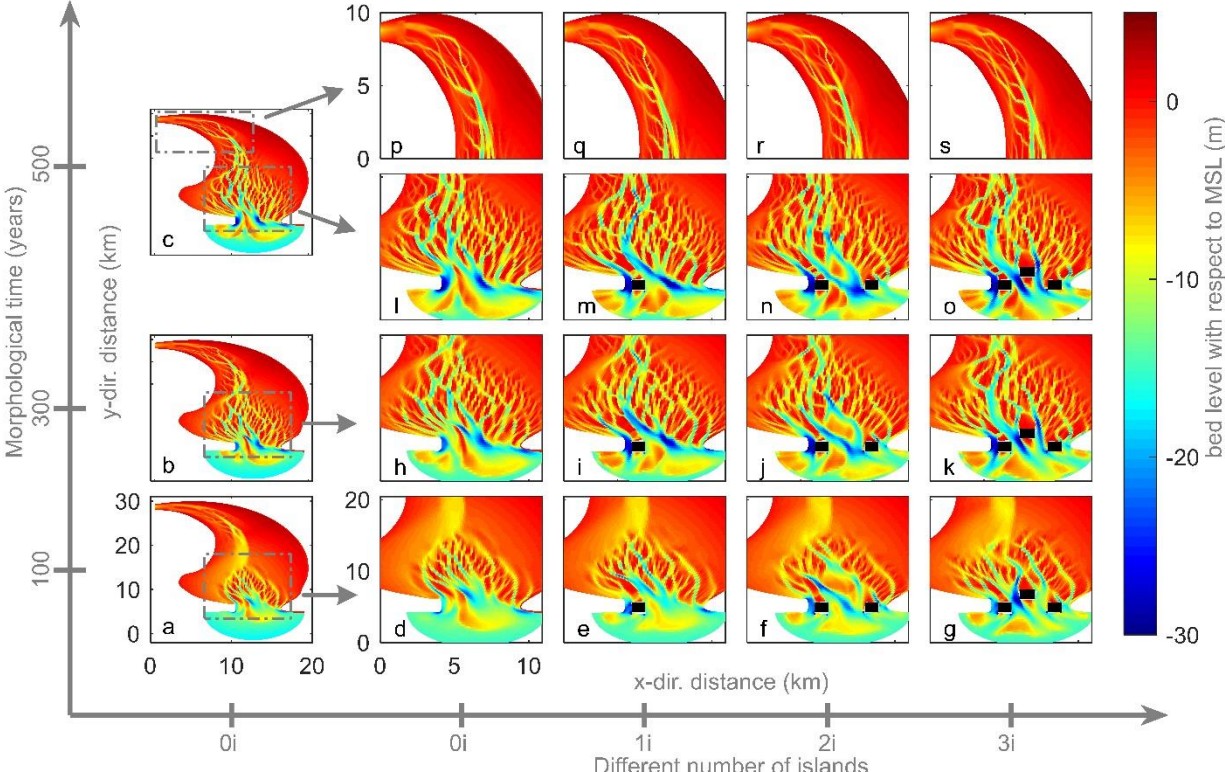

**Figure 4: Morphological evolution after 100 years (a, d-g), 300 years (b, h-k) and 500 years (c, l-s) of basin without island ("0i"), basin with one island ("1i"), basin with two islands ("2i") and basin with three islands ("3i") after the same morphological time respectively. Here we only plot part of the model domain, where the bed level changes. The black solid rectangles in subplots m, n, o, i, j, k, e, f and g represent mouth islands.**

The presence of mouth islands leads to stronger tidal currents at the inlet mouth. The initial flow field near the inlet at flood tide and ebb tide is shown in Figure 5. The existence of a mouth island divides the tidal inlet into two parts and creates another narrow tidal inlet, forming a dual-channel system. The narrower cross-section of the inlet also causes the increases of current velocity both at flood tide and ebb tide (Figure 5), leading to more sediment suspended and transported, forming a deeper channel in the inlet. On the other hand, due to the sheltering of the non-erodible island, a large back-barrier deposition is observed behind the island, where the tidal current velocity is relatively small (Figure 5). When another island is added at the right side of the inlet, the cross-sectional area of the inlet is further narrowed, which makes the current velocity further increases at the inlet (Figures 5c, 5g). For the case of three islands (3i), an even larger flow velocity is observed both at flood tide and ebb tide (Figures 5d, 5h).

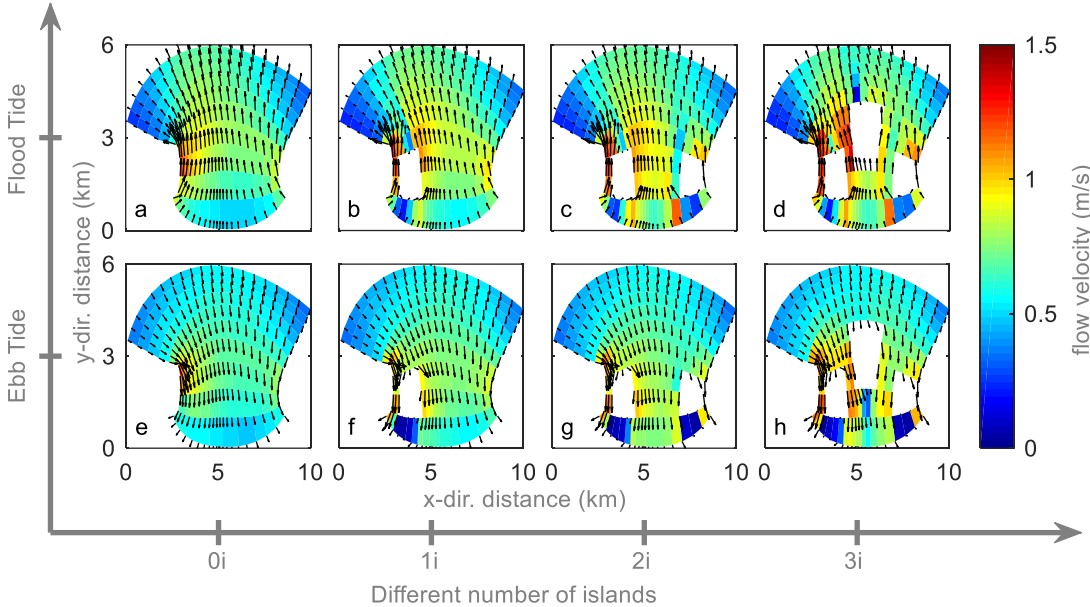

**Figure 5: Flow field near the inlet: (a-d) represent at the time of flood tide of basin without island ("0i"), basin with one island ("1i"), basin with two islands ("2i") and basin with three islands ("3i") and (e-h) represent at the time of ebb tide respectively. Arrows for flow direction and colour for depth-averaged flow velocity (m/s).**

## 3.2 Influence of the location of mouth islands

The morphological evolution of cases with different locations of mouth islands is shown in Figure 6. Tidal channel networks quickly develop in the first 100 years and gradually become stable after 300 years. However, tidal basins with mouth islands of different locations show quite different morphological patterns near the inlet but a similar pattern in the upstream estuary zone. Initially, an idealized bed elevation is defined as shown by the black dotted line in Figures 6d-6l. Three cross-sections are selected along the estuary to show the detailed morphological differences between different cases. The cross-section 1

(CS1) is far away from the tidal inlet and the river discharge is relatively small, thus the effect of hydrodynamics on morphologies at this cross-section is limited. Hence, all cases show small bed level changes and develop a similar cross-sectional bed elevation after 500 years (Figures 6d-6f, 6m). While near the tidal inlet (CS2), the tidal channels develop quickly in the first 100 years showed by large bed level changes. As the morphological evolution continues, the channel gradually develops into upper intertidal area forming a complex channel network. For the scenarios of inlet island ("IL"),

it develops a larger number tidal channels compared with the other two cases (Figures 6g, 6n), indicating that this type of mouth island can lead to tidal currents disperse into the basin. For the case of basin-side island ("BS"), a better developed channel network is found in the left side of tidal basin and the tidal channels are gradually merge together, showing a wider but a smaller number of channel network (Figure 6h, 6n). The tidal channels in the scenarios of delta-side island ("DS") are relatively shallow but there is a main channel developed at the middle of basin (Figure 6i). That's because the presence

of this type of mouth island leads to larger currents beside the island but smaller currents behind the island, thus resulting in the convergence of tidal currents entering the basin and larger erosion in the middle of tidal basin. In terms of ebb-delta area (CS3), model results also show a different morphological development in different cases. On both sides of ebb-delta, it suffers large erosion and develops tidal channels. While in the middle of ebb-delta, it shows large deposition and develops ebb-delta.

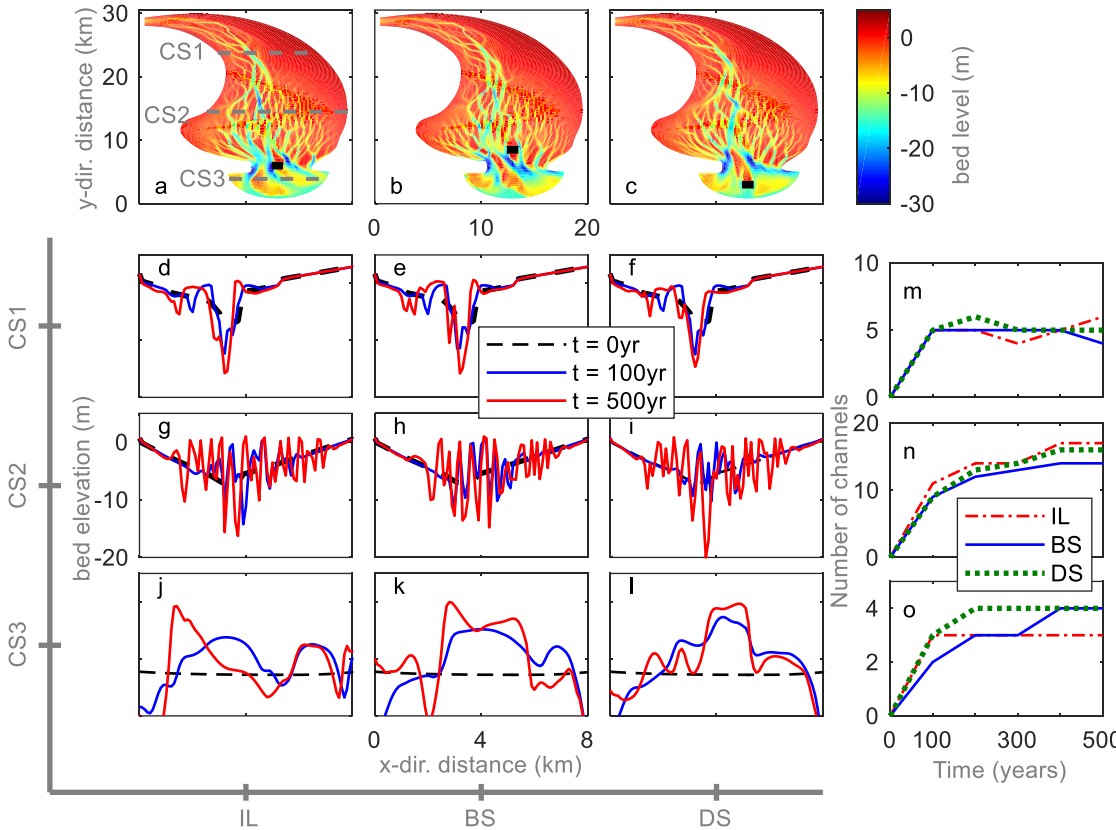

**Figure 6: Morphological evolution after 500 years (a-c) of basin with mouth island at the inlet ("IL"), at the basin-side ("BS") and at the delta-side ("DS") respectively. Figures d-l represent the temporal cross-sectional bed elevation of different cases: (d-f) cross-section 1; (g-i) cross-section 2 and (j-l) cross-section 3. Figures m-o represent the number of channels of different cross-section respectively. The black rectangle area in some figures represents the mouth island and the grey dotted lines represent the position of cross-section.**

### 3.3 Patterns of residual currents and residual sediment transport

The above morphological evolution characterised by the formation of shoals and channels is highly linked to the variation in residual tidal current and sediment transport patterns. In this section, we compare tidal residual currents in the beginning and after 300 years to illustrate the mechanisms of this type of evolution a basin (Figure 7). The residual currents are calculated by averaging the flow field over a tidal cycle, which produce residual sediment transport, leading to the morphological evolution of basin (Leonardi et al., 2013).

The presence of mouth islands leads to a higher residual current. Initially, the residual currents near the inlet are mostly landward (flood-directed) and circulating residual currents are found outside the basin (Figures 7a). When a mouth island is added, some of the tidal residual currents are directly reflected back into the sea, while others enter the inner basin through narrowed inlets with a stronger current velocity (Figures 7b). If another island is added at the right side of the inlet, the spatial distribution of residual currents is approximately symmetric and two circulating residual currents are formed behind the island (Figures 7c). As the inlet becomes much narrower, the landward residual currents become much stronger. For the case of three islands (3i), the residual currents are larger than that of the other cases, leading to a stronger residual sediment transport in the basin. In the beginning, the residual currents are relatively large with the tidal basin being morphodynamically active, so tidal flats and channels develop rapidly in the first decades. After 300 years, the residual currents decrease and the basin morphology tends to be stable (Figure 4). The formation of tidal channels and sand bars has

a significant impact on the spatial distribution of residual currents (Figures 7e–7h). The residual currents decrease to the magnitude of approximately 0.3 m/s after 300 years, indicating that hydrodynamics gradually adapt to basin morphology towards a relative equilibrium state.

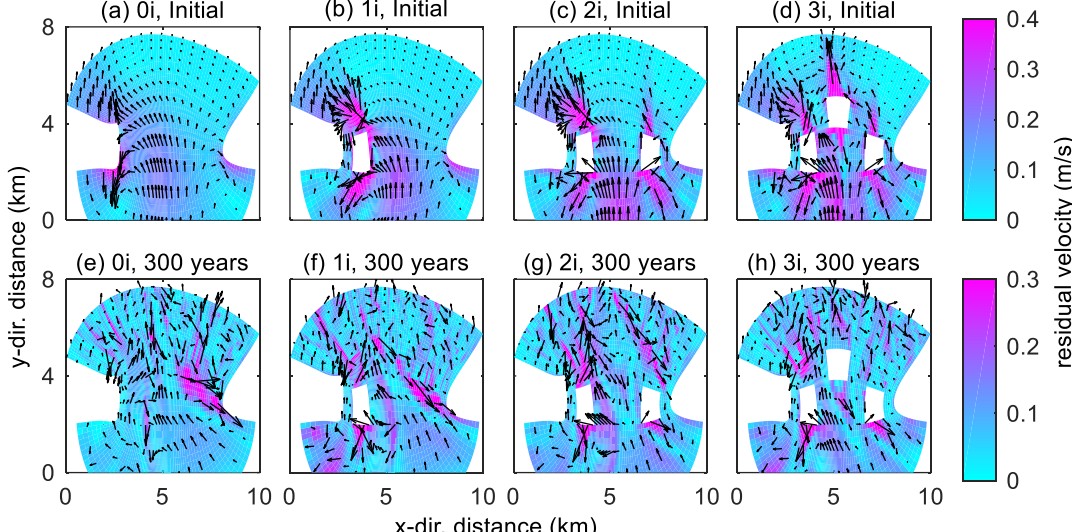

**Figure 7: Spatial distribution of the residual current (arrow vectors indicate the direction) near the tidal inlet of four different scenarios: (a-d) represent without island (0i), one island (1i), two islands (2i) and three islands (3i) at the beginning respectively. (e-h) after 300 years. The white area in the plot represents the mouth islands and the background color represent the magnitude of residual velocity.**

The presence of mouth islands enhances seaward residual sediment transport. The temporal evolution of the residual sediment pattern near the tidal inlet is shown in Figure 8. The cross section is located in the delta-side near the tidal inlet and ensure that the initial cross section area is the same for all cases (Figure 8h). In the beginning of morphological evolution, residual currents are landward (Figures 8a-8d) but there is a seaward net sediment transport is observed in the middle of tidal inlet, forming a two-way transport pattern in the tidal inlet (Figures 8a). When a mouth island is added, a seaward

residual sediment transport can be observed behind the mouth island (Figures 8b). Further increase in the number of mouth islands result in the increase of magnitude of seaward residual sediment transport (Figures 8b-d), and the maximum magnitude in the "3i"-case can reach about 0.8 m$^3$/s (Figure 8o). Since the residual sediment transport patterns control the morphological evolution of tidal basins and estuaries, the magnitude of residual sediment transport determines the rate of morphological changes (Guo et al., 2015). As shown in Figure 8o, the "3i"-case has relative higher residual sediment

transport, thus the tidal system may develop and evolve more rapidly than the scenarios with few mouth islands. In the cases of different island locations, a larger area of seaward residual sediment transport is observed when the mouth island moved further into the basin (Figure 8e-g). Model results indicate that the basin-side island tends to result in a larger residual sediment transport than the delta-side island (Figure 8q-r). After 300 years of morphodynamic development, the residual sediment transport pattern is highly affected by the developing channels and shoals and the magnitude of sediment transport

has decreased significantly, only 0.1-0.2 m$^3$/s (Figure 8p, r). As the morphological evolution continues, the magnitude of residual sediment transport continues to decrease, and it seems that the final state of the sediment transport turns to a similar pattern among all cases (Figure 8h-k), indicating that the evolving basin morphologies adapt to tidal hydrodynamics towards a morphodynamic equilibrium state. However, it is worth noting that although the residual sediment transport gradually decreases over time, it is never close to zero over the whole tidal cycle and a dynamic equilibrium state is formed (Zhou et

al., 2017).

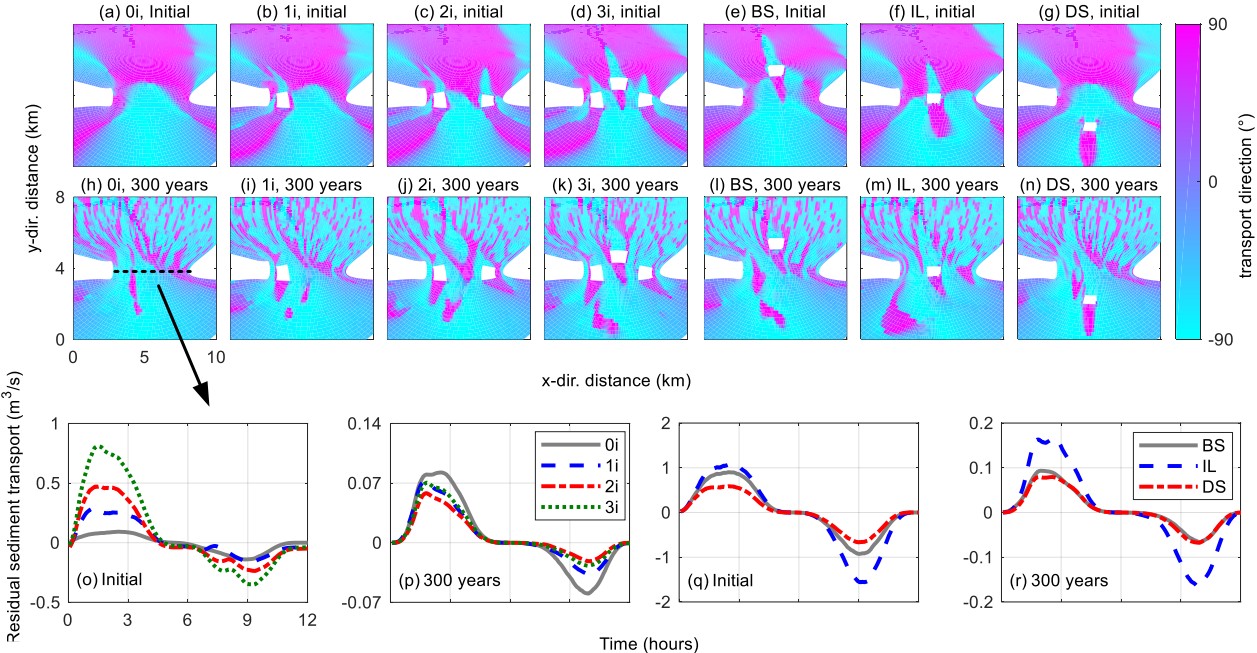

**Figure 8: Residual sediment transport pattern near the tidal inlet in the scenarios of different number of mouth islands (a-d, h-k) and different number of mouth islands (e-g, l-n) at the beginning and after 300 years respectively. The white area in the plot represents the mouth islands and the background colour represent the direction of residual sediment transport (90 degrees indicate landward transport while −90 degrees indicate seaward transport). Tidal residual sediment transport via cross-section of different scenarios at time of 0 year (o, q) and 300 years (p, r) respectively. In each plot, positive value indicates seaward transport while negative value indicates landward transport.**

### 3.4 Hypsometry curves and "P-A" relation

One useful metric that links the morphology to the hydrodynamics of tidal basins is hypsometry, which can provide information on the percentage of shoal area and channel area (Townend, 2008; Vivoni et al., 2008). The hypsometry of the inner basin for different scenarios after 100 and 500 years is shown in Figure 9, and we divide the intertidal zone and subtidal zone according to the tidal amplitude (1.2 m). Initially, a schematic profile is defined and an elevation inclined linearly toward the sea, thus the hypsometry appears to be linear (see the grey dotted line in Figure 9a). In the first 100 years, all cases show a rapid development of channels and tidal flats, indicating vertical redistribution of sediments. As the increase of mouth islands, a more pronounced development is found in the 3i-case (see the green dash-dot line in Figure 9a). Compared to the hypsometric curves at 100 years, the ones gradually move to the left side and become convex after 500 years, which indicates that shallower tidal flats and accreted shoals are developed in the basins (see dotted lines in Figure 9a).

Tidal basin with different location mouth island shows somewhat differences in hypsometric curves. Since all models use the same initial bathymetry, the initial hypsometry curve is the same as the cases of different number of mouth islands (see the grey dotted line in Figure 9b). After 100 years of development, there are less accreted flats in the basin with a delta-side mouth island. While the basin with a basin-side mouth island develops largest volume of tidal flats among the three cases (see the dash-dot line in Figure 9b). After 500 years of evolution, almost the same magnitude of shoals and tidal flats are developed in all cases (see dotted lines in Figure 9b).

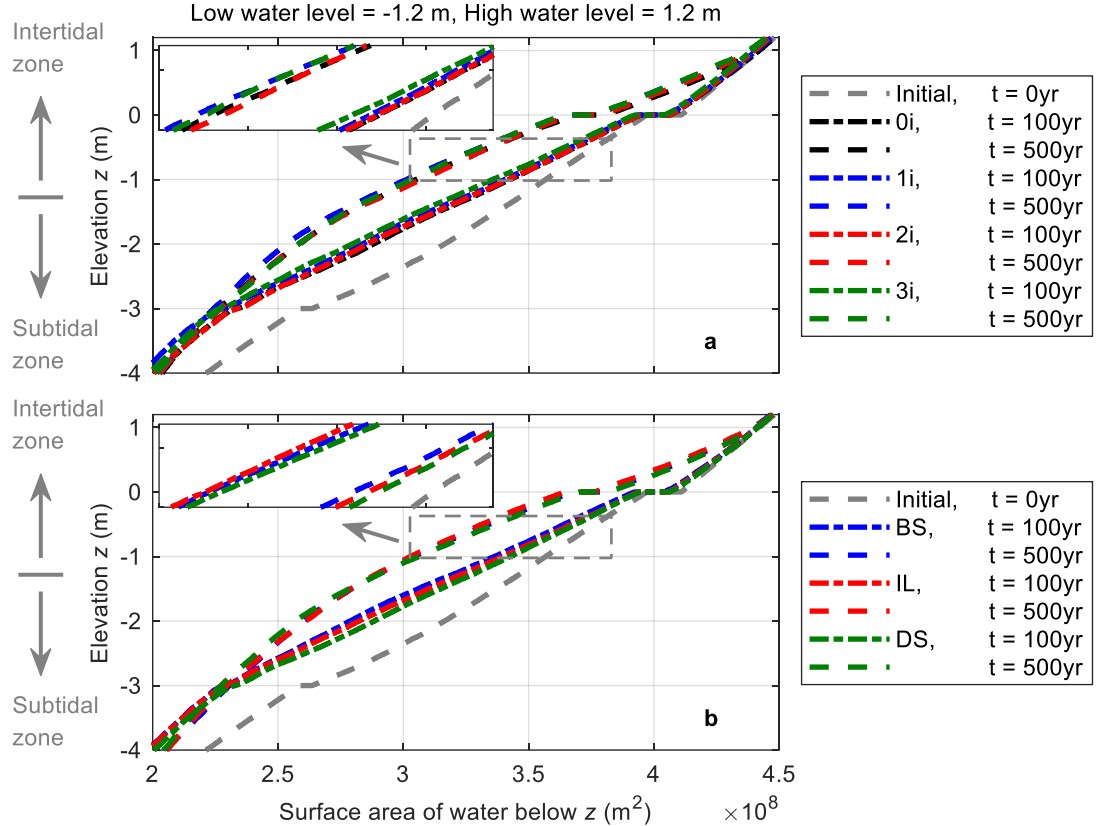

**Figure 9: Hypsometry of the tidal basin for the simulations of different scenarios: (a) the scenarios of different number of mouth islands; (b) the scenarios of different location of mouth islands after 100 and 500 years respectively. The tidal amplitude in all cases is 1.2 m.**

Another useful analysis connecting the geomorphic characteristics and hydrodynamic forces is the relationship between the cross-sectional area and the tidal prism (P-A relation) (O'brien, 1931; Jarrett, 1976; Friedrichs, 1995; Townend, 2005) The P-A relation as follows:

$$A = KP^n \tag{3}$$

where A is the cross-sectional area (m$^2$), P is tidal prism (m$^3$), K and n are fitted coefficients. The evolution of P-A relation is closely related to a number of factors, such as the hydrodynamic forces, sediment transport, and geological landform. In recent decades, many numerical studies have been conducted to explore the P-A relation of estuaries (Lanzoni and Seminara, 2002; Van Der Wegen et al., 2010) and tidal inlets [*Powell et al.*, 2006; *D'Alpaos et al.*, 2010; *Zhou et al.*, 2014]. However, few studies consider the effect of mouth islands on the morphology of basins, and our knowledge on the P-A relation of mouth basins is limited. Since islands are common landforms in the mouth of tidal basins, it is therefore of great significance to explore their influence on the P-A relation. In this study, we adopt a widely used method to calculate the tidal prism, following the studies of Savenije (2012) and Zhou et al. (2014a). Specifically, the tidal prism is calculated by the flow flux volume through a defined cross-section during the flood and ebb. In this study, the minimum width cross section is used, as shown by the black dotted line in Figure 8h.

$$P = \sum_{1}^{n} v * h * \Delta y * \Delta t \tag{4}$$

Where n is the number of grids in the cross section, $v$ is the velocity component along the inlet, $h$ is the water depth, $\Delta y$ is the grid size of the cross-section, and $\Delta t$ is the hydrodynamic time step. The variation of the tidal prisms and the inlet cross-sectional area for different mouth island scenarios is shown in Figure 10. For the scenarios of different number of

mouth island, the existence of islands reduces the cross-sectional area of the inlet, but increases the tidal current velocity, so that the initial tidal prism of the "1i"-case, the "2i"-case and the "3i"-case is larger than the "0i"-case (Figure 10a). In the first 100 years, a rapid increase is observed in the tidal prism and cross-sectional area because the tidal basin is far away

from the equilibrium state, characterized by the development of tidal flats and channels. As for different number of mouth island scenarios, the tidal prism of the "3i"-case increases fastest in the first 100 years, and it also develops the more tidal flats and channels (Figure 10a). However, after 100 years, the tidal prism begins to decrease gradually, while the cross-sectional area tends to be stable (Figure 10b). This is because the developed shoals directly decrease the accommodation space for water and thus also lead to the tidal prism decrease.

For the scenarios of different location of mouth island, model results also show that a sharp increase is observed both in the tidal prism and the cross-sectional area in the first 100 years (Figures 10c-d). The basin with a basin-side mouth island has the relatively larger tidal prism and cross-sectional area compared with the other two cases. This is because although they have the same inlet cross-sectional area, the basin-side island can further increase the current velocity entering the tidal basin. Similarly, there is a decrease of tidal prism in all cases after 100 years, but the cross-sectional area is still evolving,

which is characterized by the deepening of water depth. After 300 years, the tidal basin tends to be stable showed a gradually stable tidal prism (Figure 10c).

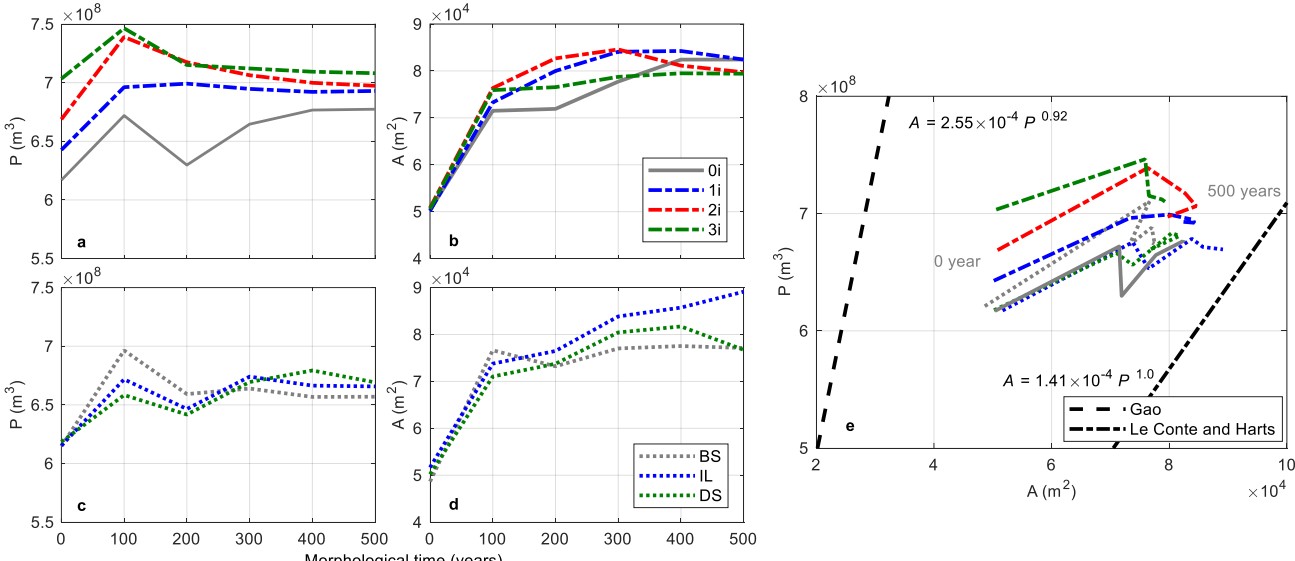

**Figure 10: The evolution of tidal prism, inlet cross-sectional area and "*P-A* relation" for the different mouth island scenarios over 500 years. (a) and (b) for the scenarios of different number of mouth islands, (c) and (d) for the scenarios of different location**
**of mouth islands.**

The tidal basins with different mouth island scenarios gradually evolved to stable morphologies (Figure 4), which is characterized by the decrease of residual current and sediment transport flux (Figure 8). Similarly, the tidal prism and inlet cross-sectional area in different scenarios tend to develop to a stable value (Figure 10a-10d). Model results show that the number and location of mouth islands affects the P-A relationship. For example, the basins with different number and

345 location of mouth islands develop different tidal prism and cross-sectional area evolution trends. Similarly, with the increase of the number of mouth islands, the tidal prisms of these tidal basins increase gradually and tend to have a similar development trend, but their cross-sectional area evolution is quite different. Figure 10e shows the simulated trajectories of P-A points at different times and compared with some of existing empirical P-A relationships (e.g., Le Conte and Harts (1905) and Gao (1988). These scattered P-A points of different mouth island basins tend to evolve toward an equilibrium

state and the tidal basins with mouth islands may take a shorter time to reach an equilibrium. Model results suggest that the tidal basins with mouth islands appear to be able to develop a larger tidal prism and cross-sectional area. As indicated by previous studies, the P-A relationship is highly affected by many factors, including site-specific and scale-dependent factors, (D'alpaos et al., 2010), hydrodynamic and sediment properties (Townend, 2005), river discharge and initial basin bathymetry (Van Maanen et al., 2013a; Zhou et al., 2014a). This study suggests that the number and location of mouth
islands as geological constraints near the tidal inlet can also play an important role on the evolutionary trends of basin morphology, thus affecting the P-A relationship.

## 4 Discussion

### 4.1 How does the mouth island affect basin morphology?

The existence of mouth islands affects the local hydrodynamics near the inlet and the sediment transport patterns, and hence
the long-term morphological evolution of tidal basins. Figure 11 shows the cross-sectional average velocity at the tidal inlet over one tidal cycle. Initially, the maximum flood and ebb tidal velocity in the case of "0i" are nearly the same (about 0.58 m/s). When a mouth island is added, the tidal current velocity increases both during the flood and the ebb. Tidal velocity increases with the increase in the number of islands (Figure 11a). With respect to different island locations, a larger flood and ebb tidal velocity is observed in the case of "BS" (Figure 11a). After 300 years, tidal velocity in the case of "0i"
decreases slightly to about 0.5 m/s, while in other cases it decreases significantly. The tidal basin with larger numbers of islands also has a higher velocity (Figure 11b). When the basin evolves to 500 years, the tidal velocity in all cases decrease only slightly, indicating that the systems were approaching morphodynamic equilibrium.

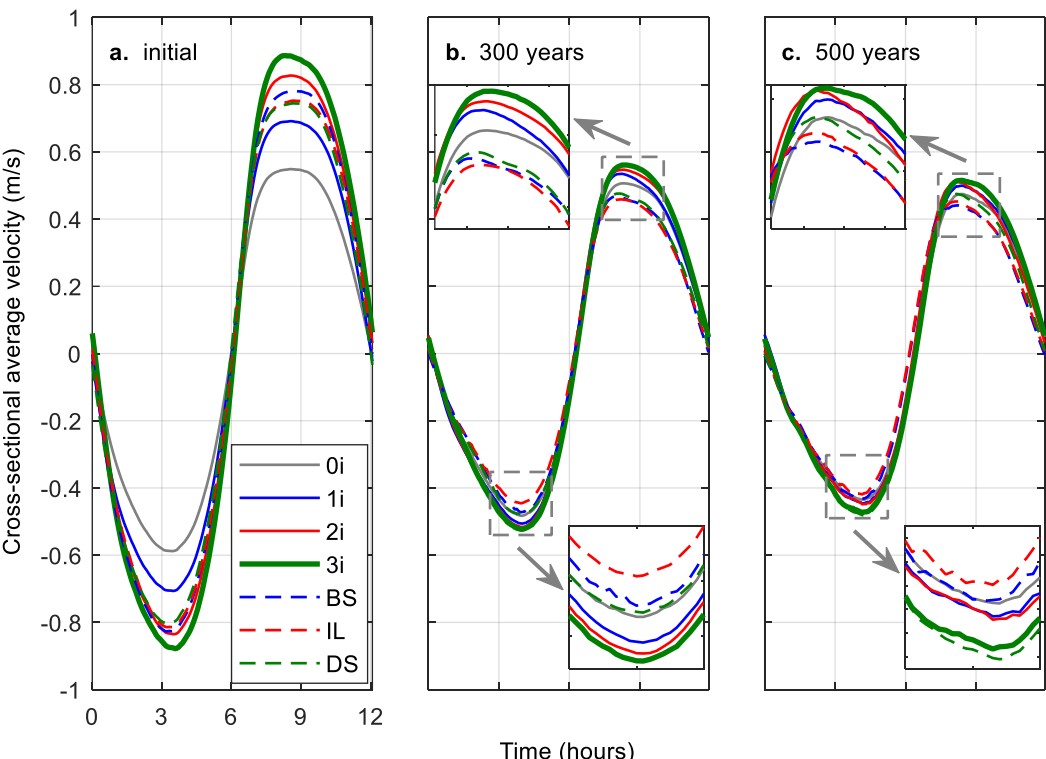

**Figure 11: The cross-sectional average velocity over a tidal cycle of different scenarios after: (a) 0 years, (b) 300 years, and (c)**
**500 years. The flood tidal velocity is positive.**

The presence of mouth islands at different locations alters the flow velocity near the tidal inlet, resulting in different sediment suspension and transport, and plays an important role in morphological evolution. Figure 12 shows the development of the tidal flat area, intertidal storage volume ($V_s$), channel volume ($V_c$) and ratio of $V_s/V_c$ and a/h of different geological conditions respectively. In this study, the tidal flat area is defined as the area between mean high water (MHW) and mean low water (MLW), $V_s$ is defined as the volume of water over the intertidal, and $V_c$ is defined as the total water volume below MLW (Friedrichs and Aubrey, 1988). In terms of tidal flat area, it is found that the tidal flat develops rapidly in the first 300 years (Figure 12a). The existence of basin-side island contributes to the development of more tidal flats in tidal basins, which has the fastest development rate. After 300 years of evolution, the development of tidal flat slows down, and the tidal basin tends to be stable gradually (see also Figure 4). But this does not mean that morphodynamic equilibrium is reached at the end of 300 years, because the intertidal storage volume and the channel volume are still developing and evolving (Figures 12b-c). In terms of intertidal storage volume, there is a gradually decrease after 300 years, indicating the tidal flat is still developing (Figure 12b). Different from the tidal flat area and the intertidal storage volume ($V_s$), the channel volume ($V_c$) shows sharply decreasing in the first 100 years and then continuously increasing. It is likely that the morphological development in the first 100 years was mainly caused by the horizontal redistribution of sediment, while the deepening of channels led to the continuous development and evolution of tidal flats and channel networks. The ratio of $V_s/V_c$ and a/h is an indicator to determine the tidal asymmetry condition of a tide-dominated system (Friedrichs and Aubrey, 1988). Overall, the a/h ratio is small (<0.3) for the three cases (Figure 12d) and the ratio between $V_s/V_c$ and a/h increases in the first 100 years and then graudally decreases. Following Friedrichs and Aubrey, [1988], a numer of modelling efforts suggest that the tidal systems were flood-dominant (a/h >0.3) or ebb-dominant (a/h<0.2), while the other key parameter $V_s/V_c$ became crucial (0.2<a/h<0.3). Due to the a/h value of the three cases is about 0.2-0.3 and the parameter $V_s/V_c$ is about 0.15-0.23, which suggests that the tidal basins are a ebb-dominated system. Model results also indicate that the effect of mouth island is limited on the tidal asymmetry.

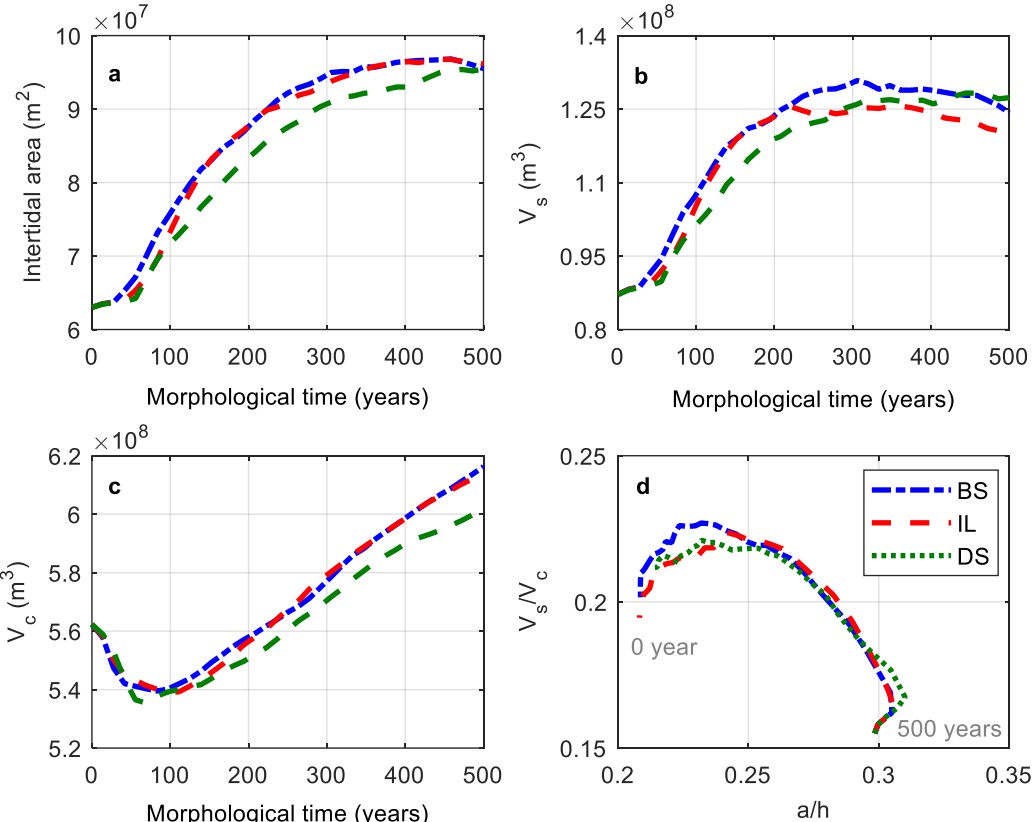

**Figure 12: Temporal variation of intertidal flat area (a), intertidal storage volume (b), channel volume (c) and ratio of $V_s/V_c$ in the inner basin of different island locations.**

The mouth islands also affect the sediment transport process and distribution. The temporal changes of cumulative erosion or sedimentation in tidal system are shown in Figure 13. We find that, all cases show sedimentation in the ebb-delta and erosion in the tidal basin (Figure 13). If mouth islands are added, the inner basins undergo more erosion with the morphological evolution, because of the presence of islands that produce larger flow velocities, resulting in more sediment transport. Accordingly, the delta shows sediment deposition and the sedimentation volume increases continuously with the increase of mouth islands (Figure 13a).

As for the scenario runs investigating the role of mouth island location, model results also show that erosion occur in the inner basin and sedimentation occur in the delta (Figure 13b). The basin with a basin-side mouth island has larger magnitude of erosion or sedimentation, indicating that more sediment suspension and transport in the basin. That is probably because the mouth island at this location can boost and increase tidal currents entering the basin, which play a determined role in sediment transport and morphological evolution. For the case of delta-side island, the sediment volume change is relatively small, which indicates that this type of mouth island can directly alter the tidal current into the basin and play a key role in local sediment transport process (Figure 13b).

In this study, it is worth noting that the residual currents are landward in the initial bathymetry, while the net sediment transport is seaward (Fig. 13). The river discharge is relatively small (50 m$^3$/s), its impact on residual current and residual sediment transport is therefore limited. A possible explanation can be provided in terms of the Stokes return flow that interacts with the tidal current generating larger residual sediment transport than residual current (Guo et al., 2014). A phase lag between the water levels and velocities induces a landward Stokes drift that causes a landward accumulation of water and momentum, resulting in a water level gradient (negative seaward) (Van Der Wegen and Roelvink, 2008; Van Der

Wegen et al., 2008). This water level gradient induces a seaward return flow (Stokes return flow), enhancing the ebb dominant and exporting character of the basin.

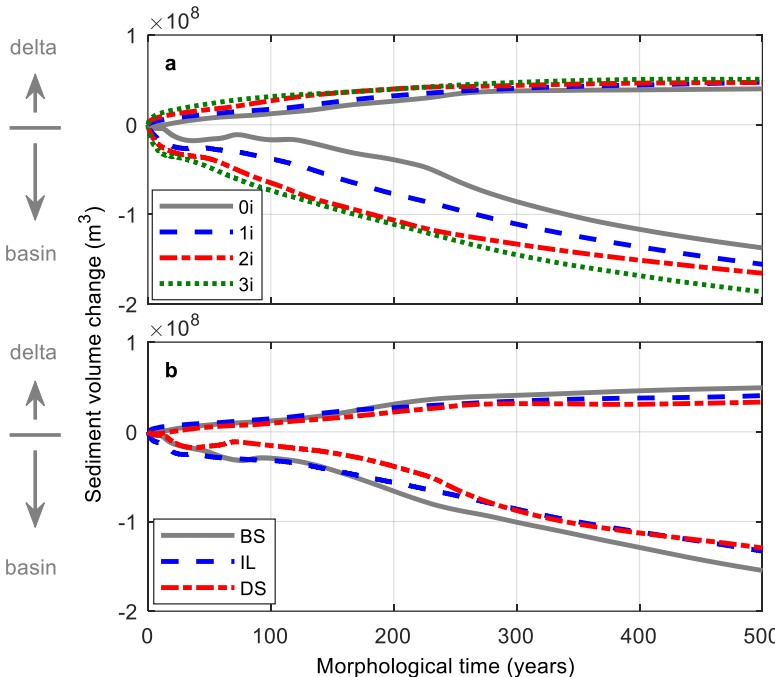

**Figure 13: Cumulative sediment volume change over time of different scenarios: (a) scenarios of different number of mouth islands, and (b) scenarios of different location of mouth islands.**

**4.2 How far is the area affected by the mouth island?**

In this section, we use a reference case ("0i") to quantify the impact of mouth islands at different locations and discuss how large the area is affected by the islands. Figure 14 shows variations of cross-sectionally averaged flow velocity and sediment transport compared to the case of zero island in the longitudinal direction.

The mouth island has a great influence on the hydrodynamic and sediment transport in local areas (e.g., tidal inlet), but the

425 variations in the scale of the whole tidal system are relatively small. In terms of flow velocity, we use the case of zero island ("0i") as a reference case and calculate the variations between other cases and reference case.

$$r_v = \frac{(v-v_{0i})}{v_{0i}} \tag{5}$$

where, $v$ is the velocity in other cases (m/s), $v_{0i}$ is the velocity of reference case ("0i"). As shown in Figure 14a, the variations mainly concentrate near the tidal inlet, and gradually decrease both in the landward and seaward directions. If

the mouth island is located at the delta-side of basin, there is a decrease in the front and back of the island and an increase on both sides of the island. The velocity in other areas is almost the same as the reference case (see the green dotted line in Figure 14a), indicating that this type of island has little impact on the flow velocity in the basin. If the mouth island is located at the tidal inlet, a greater increase of 70% is observed near the tidal inlet, but the velocity decreases sharply in the tidal inlet. In the tidal basin, it decreases by 30% at 5 km away from the tidal inlet (see the red dotted line in Figure 14a).

When the mouth island is located at the basin-side of inlet, a greater increase of 60% is observed near the island, and then gradually decreases in the landward direction (see the blue dotted line in Figure 14a). Similarly, variations are highly concentrated near the tidal inlet in terms of cross-sectionally sediment transport. We also use a coefficient ($r_s$) to quantify the differences between different cases.

$$r_s = \frac{(s - s_{0i})}{s_{0i}} \qquad\qquad (6)$$

Where, $s$ is the sediment transport in other cases (m/s), $s_{0i}$ is the sediment transport of reference case ("0i"). Comparing the different curves, we can easily find that the cross-section where the islands exist can lead to a larger sediment transport. Moreover, a sediment transport more than 3 times that of the reference case is found in the case of "IL". It is worth nothing that the relative differences in sediment transport between the cases are much larger than those in flow velocity due to the non-linear relationship between sediment transport and velocity. However, in other areas, very small variations are observed

both in the inner basin and in the outer basin (Figure 14b).

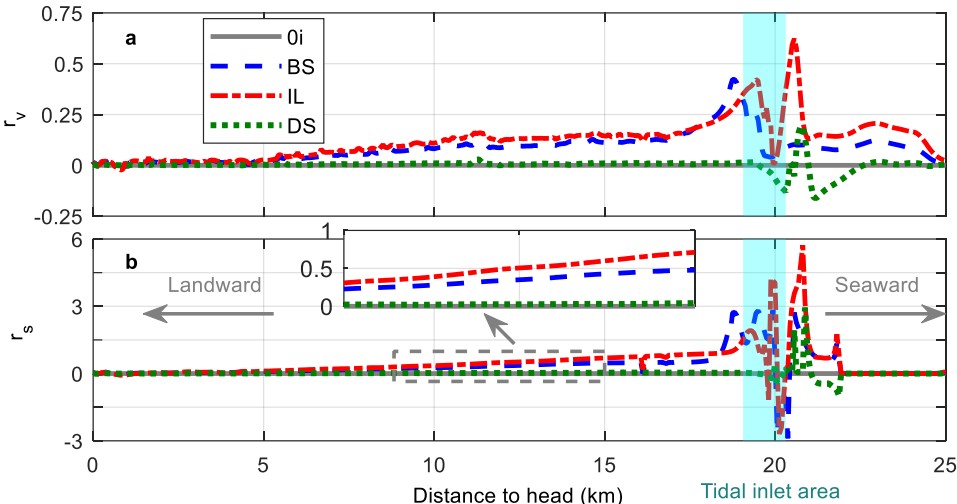

**Figure 14: Longitudinal variations of cross-sectionally averaged flow velocity (a) and sediment transport (b) compared to the case of zero island at high tide of the first year.**

**4.3 Implications for realistic tidal basins**

Even though this study is inspired by observations at a pair of adjacent tidal basins, this study has been highly simplified in order to gain direct knowledge of the role of mouth islands. Numerical experiments demonstrate that the presence of mouth islands can significantly affect the local hydrodynamics and residual sediment transport, and thus influence local channel-shoal morphology. However, in this study, all simulations result in the development of a dendritic channel network

that cannot explain the dichotomy shown in the comparison, which implies that mouth islands are only one of the determinative factors contributing to the overall shape of the basin morphology. It is therefore worth nothing other potential effects that possibly lead to the different morphology.

From a morphodynamic standpoint, initial bathymetry and tidal range play a significant role in the development of channel network and intertidal area (Dastgheib et al., 2008; Van Maanen et al., 2013a; Zhou et al., 2014a). Initial bathymetry

influences the overall sediment availability, while tidal range affects the bed level change by determining the amount of sediment can be redistributed. By comparing two tidal inlets that they are close and sharing similar tidal ranges, but we observe quite different bathymetries (Figure 1). The average water depth in Massachusetts Bay is about 6 m (Signell and Butman, 1992), which is larger than 2 m in Plymouth Bay (Gontz et al., 2013). This shows that different amounts of sediment can be redistributed in the basins, which is one of possible reasons accounting for the differences in morphology.

Besides, the sedimentary environment is also different. Massachusetts Bay is a muddy environment, while Plymouth Bay is sandy (Ford, 2010). The sediment properties are found to influence the final profile shape and vertical distribution of

sediment (Zhou et al., 2016), thus affecting the overall morphology. Also, a stronger velocity is developed at the inlet of Massachusetts Bay (Knebel et al., 1991), which may also result in the suspension and export of fine sediment. Finally, human activities have a major impact on basin morphology and since the Boston harbour is in Massachusetts Bay, waterway dredging can be one of the main factors attributed to the deeper watershed and over-deepened channel.

With the focus on the role of mouth islands, some assumptions and simplifications have been inevitably made in our numerical modelling, in order to make it easier to interpret model results and clarify potential insights. However, further research effort should be made to shed light on some of the neglected mechanisms: (1) The mouth islands are considered in the model as non-erodible and of rectangular shape, while natural islands are often slowly eroded with time and have irregular shapes. (2) The effect of wave action is excluded in the model while it may have a great influence on the morphodynamic evolution of tidal basins especially on the ebb-delta area. Nearshore waves can enhance alongshore sediment resuspension and drift, resulting in more sediment being transported to the open sea and forming larger ebb deltas (Hayes, 1980; Fagherazzi and Wiberg, 2009). (3) Sea level change is not considered in this model, while it may play a remarkable role in the morphological evolution at the centennial and millennial timescales. Particularly, some of the low islands may be submerged with sea level rise. Besides, existing studies have suggested that the sediment transport pattern may shift from exporting to importing forced by sea level rise (Dronkers et al., 1990; Van Der Wegen, 2013; Van Maanen et al., 2013b). (4) Salt-tolerant vegetation (e.g., salt marshes and mangroves) is found to play an important role on basin morphological evolution, which is not considered in this model. A number of studies have indicated that vegetation can trap and stabilize sediment by decreasing the flow velocity (Townend et al., 2016; Chen et al., 2018). On the other hand, the sedimentary environment can also help vegetation to grow, forming positive feedback between morphology and vegetation.

**5 Conclusion**

In this experiment, we numerically investigate the effects of the number and the spatial distribution of geological mouth islands on long-term evolution of a back-barrier tidal inlet system. Model results indicate that both the flow velocity and residual currents increase with the number of islands, thus enhancing the sediment transport and bed level change near the inlet. Erosion tends to occur in the tidal basin and sedimentation in the ebb-delta area, and the erosion (sedimentation) volume increases with the increase in the number of mouth islands. The spatial distribution of mouth islands is also found to be important in determining the local channel-shoal morphology of the basin and the ebb delta. If a mouth island is located at the basin side near the inlet, it tends to enhance the ebb dominance of tidal currents and hence favours the erosion of the basin while the deposition of the ebb delta. If a mouth island is located at the delta side near the inlet, it can directly hinder and divert the tidal current entering the basin and play an inhibiting role for basin morphological development. Besides, model results suggest that the number and location of mouth islands can affect the empirical relation between tidal prism and inlet cross-sectional area (the so-called P-A relation): more mouth islands result in larger tidal prisms in the basin; a basin-side island near the inlet also leads to larger tidal prisms than that of the delta-side island. The influence of mouth island on local areas (e.g., tidal inlet) is evident but very limited on the upstream estuary zone (where river starts to dominate). Overall, this study shed lights on the influence of mouth islands (which may be submerged under future sea level rise) on the long-term morphodynamic evolution of tidal basins, hence providing new insights into the evolution of these systems.

**Code and data availability**

The results are simulated using the open-source Delft3D software package (Delft3D-flow version 4.03.01) and the code is accessible through the website (https://oss.deltares.nl/web/delft3d). The input files of the default model are available at Zenodo: https://doi.org/10.5281/zenodo.5508801 (Wei, 2021).

**Author contributions**

Yizhang Wei, Zeng Zhou and Yining Chen proposed the idea for this study. Zeng Zhou, Qin Jiang, Zheng Gong and
510 Changkuan Zhang acquired the funding. Yizhang Wei built the numerical model, processed and analysed the data, and wrote the manuscript. Zeng Zhou, Yining Chen, Peng Yao, Jufei Qiu,, Ian Towend and Giovanni Coco provided logistical support and revised the manuscript. All authors were involved in reviewing and editing the manuscript, and gave final approval of the version to be published. The authors also thank the reviewers for providing detailed and constructive feedback.

**Competing interests**

The authors declare that they have no conflict of interest.

**Acknowledgements**

This study is supported the National Key Research and Development Program of China (Grant No.2018YFC0407501), the National Natural Science Foundation of China (NSFC, Grant Nos. 41976156, 51620105005), and the Natural Science
Foundation of Jiangsu Province (Grant No. BK20200077).

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
