# Peer review of "The role of geological mouth islands on the morphodynamics of back-barrier tidal basins"

_Earth Surface Dynamics, 2021_

## Referee Comment (RC1)

This paper from a very strong research group describes a study on the effects of the presence of the islands around the gorge of a tidal inlet on the morphodynamics of the tidal inlet systems using idealized numerical modelling. Basically, it is about the role of the geological constrains on the morphodynamics of tidal inlets, an interesting and important subject. The study already provides some useful insights and the paper is well written. Therefore, I support the eventual publication of the paper.

Obviously, the subject dealt by the paper is a wide one. I would consider the study described by the paper as a start for studies on the subject. Many suggestions for extending and / or improving the study can be made. I would appreciate if the authors can consider the following suggestions for revising the manuscript. I would understand if not all suggestions can be implemented before finishing the present paper, but then please consider them in the discussion section of the paper.

1. In the introduction section two pairs of nearby tidal inlets have been presented for a comparison between tidal inlets with and without islands near the inlet gorge. Can this part be extended by elaborating more on what we learn from the comparison? What are exactly the different characteristics of the geomorphology of inlets with islands from those without islands? How are the results from the comparison linked to the present modelling study?

2. Can you present something about the morphology of the Dongshan Bay, the reference tidal inlet system for the idealized modelling? Would it be possible to make a comparison between the model results and the real morphology of this bay? Even for idealized modelling study I think it is important to present some validation of the used model.

3. It seems to me that the major effect of the islands in the idealized model is narrowing the gorge of the inlet. Therefore, please discuss on what really matters, the varying width at the inlet gorge or the number of islands?

4. The model results show that sediment export takes place in all cases. Can you please discuss on the mechanism(s) causing this seawards residual sediment transport? Is this due to the residual flow velocity caused by the river discharge and the flow compensating Stoke's drift?

5. More detailed, at what time are the flow velocity and sediment transport presented in Fig.14? Please consider changing the scales of the vertical axes of pictures a and c. Picture a does not show any differences between the four cases seemingly in in contradiction with the results presented in e.g. Fig.13. The relative differences in sediment transport between the cases should be much larger than those in flow velocity because of the non-linear relationship between sediment transport and velocity. However, picture c does not show this, most likely because of the used scale of the vertical axis.

Details:

| Line | remark |
|------|--------|
| 27 | I would remove "empirical" |
| 39 | I think that you mean "anthropogenic" instead of "anthropologic" |
| 93 | Replace "under" by "with"? |
| 138-139 | Is it not prescribed that the bed level is not changing? |
| 143-145 | "which suggests"? I cannot follow the reasoning. |
| 151 | "the shape of rectangular prism"? |

| | |
|---|---|
| 188 | Replace "higher" by "stronger"? |
| 189 | Replace "show" by "shown". |
| 195 | "continue"? |
| 380-381, | What do you mean by "horizontal" and "vertical" redistribution? In the model only horizontal sediment transport (from one grid cell to another) is simulated. |
| 393 | Remove "as"? |
| 410 | "integrated averaged"? Consider removing "integrated". |
| 468-469 | I am not quite sure if you can claim "hence providing … systems". No discussion is on this is provided in the manuscript. |

---

## Author Response (AR1)

**Reply to Reviewers**

Manuscript ID esurf-2021-46 entitled "The role of geological mouth islands on the morphodynamics of back-barrier tidal basins"

*Authored by:* Yizhang Wei, Yining Chen, Jufei Qiu, Zeng Zhou, Peng Yao, Qin Jiang, Zheng Gong, Giovanni Coco, Ian Townend and Changkuan Zhang

*Date of initial submission:* 02-June-2021

*Date of decision email sent:* 25- September-2021

**Note to the Editor and Reviewers:**

The comments and suggestions of the Editor and the reviewers are copied in *italic grey font*. The reply to each comment by the reviewers is written in normal font and appears just after the original comment or question.

We wish to thank the editor and reviewers for their valuable comments and suggestions that have resulted in a more insightful manuscript.

**Reviewer No. 1**

*This paper from a very strong research group describes a study on the effects of the presence of the islands around the gorge of a tidal inlet on the morphodynamics of the tidal inlet systems using idealized numerical modelling. Basically, it is about the role of the geological constrains on the morphodynamics of tidal inlets, an interesting and important subject. The study already provides some useful insights and the paper is well written. Therefore, I support the eventual publication of the paper. Obviously, the subject dealt by the paper is a wide one. I would consider the study described by the paper as a start for studies on the subject. Many suggestions for extending and / or improving the study can be made. I would appreciate if the authors can consider the following suggestions for revising the manuscript. I would understand if not all suggestions can be implemented before finishing the present paper, but then please consider them in the discussion section of the paper.*

Reply: We wish to thank the reviewer for providing very constructive and detailed comments. We have addressed the comments carefully and merged our responses into the revised manuscript.

*1. In the introduction section two pairs of nearby tidal inlets have been presented for a comparison between tidal inlets with and without islands near the inlet gorge. Can this part be extended by elaborating more on what we learn from the comparison? What are exactly the different characteristics of the geomorphology of inlets with islands from those without islands? How are the results from the comparison linked to the present modelling study?*

Reply: We fully agree with the reviewer that it is necessary to link the comparison more closely with this study. The relevant content has been added to the introduction section and discussion.

Two pairs of nearby tidal inlets have some similar characteristics: (1) They are both semi-enclosed bays characterized by a large area inside the bay and a narrow tidal inlet, which can effectively reduce waves; (2) Forced by similar tidal currents and small river discharge (Jiang and Meng, 2008).

However, some characteristics are still site-specific: (1) Sediment composition. Massachusetts Bay is a muddy environment, while Plymouth Bay is Sandy (Ford, 2010). The difference of sediment composition may be one of the reasons for the formation of different morphologies. (2) Local hydrodynamics. Sediment composition may also be the result of the long-term interaction between hydrodynamic and sediment transport (Zhou et al., 2016a). Geomorphology is highly related to the

local hydrodynamics (Van Der Wegen and Roelvink, 2008; Coco et al., 2013), which determines the sediment transport and trigger morphological changes. By comparing two pairs of inlets, although they are close, the local hydrodynamic is somewhat different. Normally, a larger velocity is formed at the narrower tidal inlet. The inlet of Massachusetts Bay is wider, but the current near the inlet is still stronger (Knebel et al., 1991). In addition to the varying width of tidal inlets, there is also an obvious geomorphic difference driven by the presence of islands and its effect on morphodynamic processes has not been studied systematically. From this point of view, cases of different numbers of islands are designed to investigate the effect of varying inlet widths narrowed by mouth islands on morphological evolution.

*2. Can you present something about the morphology of the Dongshan Bay, the reference tidal inlet system for the idealized modelling? Would it be possible to make a comparison between the model results and the real morphology of this bay? Even for idealized modelling study I think it is important to present some validation of the used model.*

Reply: To start with, we think it is useful to recall the objectives of this present study as already introduced in the initial submission. This is a schematised modelling study with the Dongshan Bay as a reference site, aiming to provide some physical insights into the effect of mouth islands on morphodynamic evolution. Our modelling study focuses on the physical mechanisms underlying a phenomenon rather than exactly trying to reproduce a set of observations. Choosing Dongshan Bay as the reference site is useful because there are some basic data (e.g., tidal range, river discharge and major sediment type) that allow us to model.

For the interest of the reviewer, we have made a general comparison between the model results and the real morphology of Dongshan Bay (Figure R1). Model results show that the morphology is overall consistent in terms of pattern formation, and the trend of morphological evolution is also consistent with the real configuration (Liang et al., 2016), even though our model predicts a more intricate network. However, there are a number of model simplifications (e.g., our simplified tidal forcing or uniform grain size for the whole domain) and local factors (e.g., human activities) that directly affect the morphological changes. Therefore, it is difficult to really reproduce the channel-shoal morphology based on this idealized model, which is not the focus of this study and this figure would not add to the paper.

[Figure]

Figure R1 Comparison of real morphology of Dongshan Bay and model results

*3. It seems to me that the major effect of the islands in the idealized model is narrowing the gorge of the inlet. Therefore, please discuss on what really matters, the varying width at the inlet gorge or the number of islands?*

Reply: The reviewer is correct that the effects of varying widths and different numbers of islands play similar roles in term of converging flow and enhancing flow velocity.

However, the existence of islands can also exert additional influences on local tidal hydrodynamic when they are formed in different locations. Due to the obstruction of the island and the fast velocity on both sides of the island, a velocity gradient is formed between the channels and behind the island, resulting in the formation of eddies. The eddies generated by strong tidal flow past islands, have an important role in natural coastal protection, since they cause waves to refract and dissipate tidal energy (Neill et al., 2012). This provides a sedimentary environment behind the island, forming a back-barrier deposition and obviously affecting the local channel-shoal morphology of tidal basin.

Furthermore, different from a single inlet system, the existence of different numbers of islands can divide one tidal inlet into several tidal channels of varying width and velocities, affecting the local tidal asymmetry. We analyse tidal asymmetry at four locations near the tidal inlet under different cases. Model results show that a different tidal asymmetry occurs in two tidal channels of different width (Figure R2 b). The narrower tidal channel is flood-dominated while the wider one is ebb-dominated. The possible reason may be that the velocity gradient between two tidal channels leads to the tidal currents in the narrower tidal inlet flow into the wider tidal inlet, weakening its flood tide. Overall, we think the role of islands is broader than just narrowing the width of tidal inlet, since the whole hydrodynamic field can be affected.

[Figure]

Figure R2 Initial tidal asymmetry of different observation points near tidal inlet of different cases: (a) 0i-case; (b) IL-case; (c) BS-case; and (d) DS-case. In each figure, small arrows represent the direction of tidal currents, while solid arrows represent tidal asymmetry.

*4. The model results show that sediment export takes place in all cases. Can you please discuss on the mechanism(s) causing this seaward residual sediment transport? Is this due to the residual flow velocity caused by the river discharge and the flow compensating Stoke's drift?*

Reply: Yes, the reviewer is correct that the export of sediment is probably due to the Stokes return flow. In this study, the river discharge is relatively small (50 m³/s), so its impact on residual current and residual sediment transport is limited. A phase lag between the water levels and velocities induces a landward Stokes drift that causes a landward accumulation of water and momentum, resulting in a water level gradient (negative seaward) (Van Der Wegen et al., 2008; Van Der Wegen and Roelvink, 2008). This water level gradient induces a seaward return flow (Stokes return flow), enhancing the ebb dominant and exporting character of the basin. In this study, it is worth noting that the residual currents are landward in the initial bathymetry, while the net residual sediment is seaward (Fig. 7). A possible explanation can be provided in terms of the Stokes return flow that interacts with the tidal current generating larger residual sediment transport than residual current (Guo et al., 2014).

*5. More detailed, at what time are the flow velocity and sediment transport presented in Fig.14? Please consider changing the scales of the vertical axes of pictures a and c. Picture a does not show any differences between the four cases seemingly in in contradiction with the results presented in e.g. Fig.13. The relative differences in sediment transport between the cases should be much larger than those in flow velocity because of the non-linear relationship between sediment transport and velocity. However, picture c does not show this, most likely because of the used scale of the vertical axis.*

Reply: Very good suggestion. The time in the Fig. 14 is the result at high tide of the initial year. We have deleted figure a in Fig. 14 since they have little difference. We have also adjusted the vertical axis of Fig. 14 and zoom in locally, as shown in the following figure. It can be seen from the enlarged figure that the relative difference in sediment transport is indeed greater than those in flow velocity.

[Figure]

Figure 14 Longitudinal variations of cross-sectionally averaged flow velocity (a) and sediment transport (b) compared to the case of zero island at high tide of the first year.

*Line 27 – I would remove "empirical"*

Reply: Agreed.

*Line 39 – I think that you mean "anthropogenic" instead of "anthropologic"*

Reply: Agreed and modified accordingly.

*Line 93 – Replace "under" by "with"?*

Reply: Agreed and changed.

*Line 138-139 – Is it not prescribed that the bed level is not changing?*

Reply: Yes, in this way to ensure that the boundary bed level remains unchanged.

*Line 143-145 – "which suggests"? I cannot follow the reasoning.*

Reply: Following the comment, we think the reason is really a bit of a stretch, so we remove it all.

*Line 151 – "the shape of rectangular prism"?*

Reply: Sorry, "prism" should be removed. What I want to indicate is a rectangle shape.

*Line 188 – Replace "higher" by "stronger"?*

Reply: Agreed and changed.

*Line 189 – Replace "show" by "shown".*

Reply: Agreed and changed.

*Line 195 – "continue"?*

Reply: Sorry, I don't understand this comment.

*Line 380-381 – What do you mean by "horizontal" and "vertical" redistribution? In the model only horizontal sediment transport (from one grid cell to another) is simulated.*

Reply: Sorry, this may be a wrong expression. What I want to indicate is that "horizontal" means the sediment moves from one grid to another, and "vertical" means that the tidal channel is gradually deepening, but the shape and size of channel network change little.

*Line 393 – Remove "as"?*

Reply: Agreed and modified accordingly.

*Line 410 – "integrated averaged"? Consider removing "integrated".*

Reply: Agreed and changed.

*Line 468-469 – I am not quite sure if you can claim "hence providing … systems". No discussion is on this is provided in the manuscript.*

Reply: Agreed. We have revised it to the following sentence:

Overall, this study shed lights on the influence of mouth islands (which may be submerged under future sea level rise) on the long-term morphodynamic evolution of tidal basins, hence providing new insights into the evolution of these systems.

**Reviewer No. 2**

*Wei et al. design a number of numerical modeling simulations using Delft 3D to quantify the effect of islands near inlets on the hydrodynamics and morphology of bay environments. The research topic is interesting and of societal importance. Their results are interesting and well explained in general.*

Reply: We wish to thank the reviewer for providing very detailed comments and suggestions. We have addressed the comments carefully and clarified many statements.

*The first figure the authors include in the paper illustrates the potential effects that islands can potentially play on the hypsometry of bays. Those without islands seem to develop more shoals and a more distinct tidal channel network than those with islands. This is a very interesting observation, but the authors do not seem to be able to capture this dichotomy in their simulations. I would suggest the authors expand the discussion and mention additional factors that could explain these differences.*

*Maybe additional model runs in a future paper? Are there alternative geometric arrangements for the islands that have not been included that could potentially lead to larger differences in bay morphology? Is there a difference in terms of dredging between bays?*

Reply: We fully agree with the reviewer that it is necessary to expand the discussion to explain the differences. We have added some more physics-based discussion in the revised manuscript, as follows:

"4.3 Implications for realistic tidal basins

[revised manuscript text omitted]

*Line 17 – …numerically explore…*

Reply: Agreed.

*Line 62 – …the pathway and sediment distribution are different…*

Reply: Agreed and modified accordingly.

*Line 76 – …have developed…*

Reply: Agreed and changed.

*Line 77 – …As showed in Figure 1…*

Reply: Agreed and modified.

*Line 78 – In contrast?*

Reply: Agreed and modified.

*Line 95 – …as a reference basin size…*

Reply: Agreed and modified.

*Line 106 – as follows:*

Reply: Agreed and changed.

*Line 109 – … is the relative density (remove and) …*

Reply: Agreed and changed accordingly.

*Line 116 – I suggest the authors better describe the sensitivity tests.*

Reply: Agreed. We have provided more details on sensitivity tests in the revised manuscript, as follows:

"Some sensitivity tests with varying MF values are performed in order to select the MF value. Specifically, it is necessary to ensure that the increased bed elevation in each time-step is small enough relative to the water depth, so that the hydrodynamic process in the next time step is not significantly different from the morphological factor of application 1 (Ranasinghe et al., 2011; Van Der Wegen and Roelvink, 2012). In this way, on the basis of ensuring the calculation accuracy, the MF value is selected as 50 to reduce the computational cost."

*Line 121 – For the initial basin bathymetry an idealized …*

Reply: Agreed and modified.

*Line 142 – Islands can potentially be submerged or even disappear…*

Reply: Agreed and modified accordingly.

*Line 189 – …is shown…*

Reply: Agreed and changed.

*Line 205 – …in Figures …*

Reply: Agreed and modified.

*Line 212 – …slender tidal channels…?*

Reply: We have removed the words in the revised manuscript. What we want to describe is that the inlet-island case develops a larger number of tidal channels compared with the other two cases.

*Line 218 – …resulting in the convergence of tidal currents entering …*

Reply: Agreed and modified accordingly.

*Line 233 – … evolution of a basin*

Reply: Agreed and changed.

*Line 241 – …with the tidal basin…*

Reply: Agreed and changed.

*Line 245 – Initially the residual currents are 0.4m/s. Not that significant of a decrease?*

Reply: Yes, the reviewer is correct that the maximum magnitude of residual currents after 300 years only decreases to 0.3 m/s. However, they only develop in tidal channels, and the residual currents in some shallow shoal tidal flats is smaller comparing to the initial stage. Therefore, we can only point out that the residual currents have decreased, but not significantly.

*Line 251 – …of the residual sediment…*

Reply: Agreed and modified.

*Line 268 – …to a similar pattern …*

Reply: Agreed and changed.

*Line 301 – I would mention early in this section what cross-sectional area the authors refer to. Equation (4) – I suggest the authors include the limits of the summatory. Is it a function of both x and y?*

Reply: Thanks for the good suggestion. We have added some text to the revised manuscript to indicate the cross-sectional area we used. The cross section we used in this study is located in the tidal inlet (see also Fig. 8h), which has a minimum width. Besides, we have also added the limits of the summation in the Equation (4). It is a function along the inlet. The sum of the along-inlet velocity component of the whole cross section is used to calculate the tidal prism.

*Line 330 – …there is a decrease…*

Reply: Agreed and modified.

*Line 374 – …the tidal flat slows…*

Reply: Agreed and modified.

*Line 394 – …inner basins undergo more erosion…*

Reply: Agreed and modified.

*Line 401 – a key role?*

Reply: We agree that the delta-side island can affect local hydrodynamics, and to some extent, local sediment transport and morphological change. However, by comparing to the cases of islands in other locations, the sediment volume change is relatively small (Fig. 13). Moreover, the cross-sectional averaged velocity and sediment flux are similar to the case without islands (Fig. 14). Therefore, we think that the effect of delta-side island on the morphodynamic process is less than that of islands in other locations.

*Line 439 – … will shed light on…*

Reply: Agreed and modified.

*Line 445 – centennial*

Reply: Agreed and changed.

---

## Referee Report (RR1)

Comment on "The role of geological mouth islands on the morphodynamics of back-barrier tidal basins"

The work by Wei et al. is an intriguing numerical study that seeks to quantify how the number and distribution of inlet mouth islands impacts the morphologic evolution of semi-enclosed coastal basins. The results of this investigation easily spur multiple investigations for future study, and it is evident that work presented here could form the foundation of a fruitful line of research. As such, I support the publication of this paper, while acknowledging that some editing is needed to clean up grammar/syntax and add additional context to underscore the importance of the results. To this end, please see my comments below. Please note that, with regards to grammar, I have not flagged every instance of awkward wording and instead focus mostly on language that could impact comprehensibility.

General Notes:
The authors should consider being more upfront about the fact that the model uses 'rocky' fixed-dimension islands in their model experiments. This detail is not apparent until section 2.3, although it is hinted at in the abstract ("geological constraints"). Stating this clearly in the abstract will allow other researchers to connect with this topic more specifically. Additionally, the Introduction switches between citations of geologically constrained and unconstrained systems, which is somewhat confusing when presented with model investigations that focus only on rocky mouth islands. I think this confusion could be eliminated by concentrating more on embayed and drowned coasts in the Introduction, with references to sandy/alluvial-type systems as points of comparison and to make more general observations about how mouth islands have broadly been observed to function. To be clear, I am not proposing a total teardown and rebuild of the Introduction, but more of a rearrangement and rephrasing to present observations in a way that could better support the specific investigations undertaken in this study. This may also make it easier to project future work in the Discussion, simultaneously giving the reader an early understanding about the limitations of this study while suggesting the importance of this work for subsequent investigation.

Line Edits and Considerations:
Line 22-25. 'Hindering role' in this sentence can probably removed to make the latter point in this sentence easier to understand. Recommend changing the dependent clause to "[…], while the presence of delta-side islands may increase relative sediment deposition in the basin."—or something similar, assuming I have captured the concept correctly.

Line 35: The last part of this sentence should read either 'these types of coastal zones' or 'this type of coastal zone' to keep consistent with pluralization or lack thereof.

Line 42: Consider swapping 'changed landforms' for just 'landforms' to better convey that this is a two-way exchange of influence.

Lines 44-46: The sentence describing the finding of Otvos (1981) is ambiguous. Without looking to the reference, my first thought reading this line is that it is possibly referring to sediment being liberated from an antecedent high to begin early barrier formation during sea-level rise. Please clarify.

Line 48: The 'and' here is being used as a conjunction. Add a comma after 'systems'.

Line 49: Consider splitting this sentence after 'tool'. 'Virtual' is also implied by numeral modeling, so not needed.

Lines 83-85: The Massachusetts coast is heavily glacially-influenced, and bay islands are a mix of drowned uplands, glacial deposits, and alluvial features. To make the last part of the sentence more broadly accurate, could probably change to […], which are probably formed from the drowning of topographic highs during sea level rise in the post-glacial period."

Lines 91-94: When setting up the specific research questions, note that the number of islands is also being evaluated.

Lines 143-144: Merge with previous paragraph.

Lines 152-155: Suggest dropping 'Besides' from beginning of sentence. This line may also work better as the final sentence of the paragraph since it is a projection for future work, e.g. "In all cases, the same initial bathymetry is adopted so that the model results can be compared. Additionally, islands in this initial study are non-erodible (rocky) and square (1 km x 1km). In the future, different sizes and shapes of islands will be investigated to determine how these parameters impact morphological outcomes."

Line 165: Need a conjunction to link thoughts about where currents occur, something like… "Morphological evolution first occurs in the mouth zone where tidal currents are strongest, as the well as the river input zone due to fluvial input."

Line 173: Add comma after 'therein'.

Line 182: Need an article before 'larger spatial scale'. Suggest 'a larger spatial scale'.

Line 183: Consider changing 'And more erosion occurs' to 'Also, more erosion occurs' or something similar.

Lines 184-185: Slight phrasing corrections needed here. Suggesting the following: "Meanwhile, in the upstream zone, small differences are observed between the four cases, indicating that hydrodynamic effects on this area are relatively limited."

Figure 4: In the future, consider using a color palette that is more accessible to readers with color deficiencies. Again, not an issue here, but something to think about for the next study.

Line 194: I think "leading to more sediment suspended and transported, and forming a deeper inlet channel" could be rephrased more simply as just "leading to more suspended sediment transport and forming deeper inlet channels".

Lines 191-199: Reading this paragraph reminds of how much this setup mimics the processes that occur around bridge pilings, which seems to be what the modeled rocky islands act like. Could be a thought worth mentioning in the discussion.

Lines 213-214: Slight phrasing corrections needed here. Suggesting the following: "As the morphological evolution continues, the channel gradually develops into the upper intertidal area and forms a complex channel network."

Line 214: Considering flipping the word order of 'scenarios of inlet island ("IL")' to 'inlet island scenarios'. This reordering would apply to similar phrasing throughout the paragraph.

Line 219: It is generally inadvisable to use contractions in a professional paper. Please change 'that's' to, in this case, 'This is'.

Lines 224-225: Not sure what this is trying to say. Based on the previous sentence, this should say something about how morphology differs at CS3 with longitudinal placement of the island, but as currently written there's just a broad statement about where channels and deposition generally develop. Please elaborate further.

Line 234: "evolution a basin" should be "basin evolution".

Lines 248 to 249: In the final sentence of this paragraph, I think the last line should read "[…] indicating that hydrodynamics gradually adapt to basin morphology and a relative equilibrium state." Please confirm if this is the correct meaning.

Figure 8: This is my favorite figure in the paper!

Line 291: "the ones gradually move the left side and become convex after 500 years" could be reworded to be more technically accurate. Perhaps something like "modeled hypsometric curves at 500 years become noticeably convex".

Lines 294-299: Check the wording on this paragraph—the topic sentence, especially, does not really explain the result. If I understand correctly, there are really only two points here: [1] During basin morphological evolution, the area of tidal flats grows slightly slower under the delta-side scenario and slightly faster under the basin-side scenario. [2] After 500 years, the magnitude of shoals and flats developed under all cases are similar. These two thoughts can probably be appended to the previous paragraph.

Figure 9: Both the 100-yr and 500-yr curves being dashed is slightly confusing to look at. Consider either making one set solid lines, or alternatively, changing the color shading between the sets. I am thinking something along the lines of dark blue and light blue for BS, dark green and light green for DS, etc.

Lines 335-336: Suggest changing "the tidal basin tends to be stable showed a gradually stable tidal prism" to "the tidal basin tends to be stable, as shown by an increasingly stable tidal prism."—or something similar.

Lines 405-407: The second part of the last sentence in this paragraph does not really add anything to the discussion here. In light of the previous sentences in said paragraph, it reads almost contradictory, although I can see that is not the intent. Consider removing and/or elaborating.

Section 4.2: This section could be better served in section 3, as it is effectively another result and could be easily discussed in the context of section 4.1. Consider rearranging—I think this would be fairly easy to implement and would improve the flow of the Discussion.

Line 451: In this first line of this section, remind the reader again that this is referring specifically to the basins in Massachusetts.

Lines 451 to 469: I think it is possible these two paragraphs can be combined. The theme here is that the study does not fully capture the differences in the observed basins, which is probably a result of other geological constraints (particularly, initial bathymetry) that were not fully explored in this investigation. This could be used to set up a future sensitivity study to compare the relative magnitudes of morphological forcing from initial bathymetry and mouth island presence/placement.

Lines 474-475: "while natural islands are often slowly eroded with time"—for the case of rocky islands specifically? Sandy mouth islands can be completely destroyed and reformed over decadal timescales. Might be worth adding some additional specificity.

Section 4.3: Overall, in this section, I was expecting more focus on future work, since the authors mention in section 2.3 that they have designs for more study. In general, I feel this like section could be more optimistic about the model results—this is a topic that is largely unexplored, and the fact that the basins which inspired the study have some differences that are not easily explained speaks to the richness of possible investigations in this research area.

Conclusions: Depending on if suggestions for more future work discussion/speculation are considered in section 4.3, consider adding a corresponding line or two here.

---

## Author Response (AR2)

**Reply to Reviewers**

Manuscript ID esurf-2021-46 entitled "The role of geological mouth islands on the morphodynamics of back-barrier tidal basins"

*Authored by:* Yizhang Wei, Yining Chen, Jufei Qiu, Zeng Zhou, Peng Yao, Qin Jiang, Zheng Gong, Giovanni Coco, Ian Townend and Changkuan Zhang

*Date of initial submission:* 02-June-2021

**Note to the Editor and Reviewers:**

The comments and suggestions of the Editor and the reviewers are copied in *italic grey font*. The reply to each comment by the reviewers is written in normal font and appears just after the original comment or question. Thanks to the editors and reviewers for the constructive comments and useful suggestions, which has significantly raised the quality of the manuscript.

**Review Editor**

*Following the recommendation of both reviewers, I recommend the paper to be published after the authors address their minor comments.*

Reply: We wish to thank the editors for handling our submission and offering us an opportunity to be published. Each suggested revision and comment, brought forward by the reviewers is accurately incorporated and considered.

**Reviewer #1**

*The authors adequately responded to all my comments except one: "Line 195 – "continue"?". Apparently, I was not clear enough. I was only wondering if "continue" is the right word here (To my understanding "continue" suggests a process in time). Among the authors, there are some native English-speaking senior experts, so I should not worry about this.*

Reply: We gratefully appreciate the time and effort that the reviewer has dedicated to providing valuable feedback on our manuscript. We think all the comments are insightful and substantially help to result in a much better manuscript. After careful consideration, we have replaced "continue" with "further" in the revised manuscript.

**Reviewer #3**

*The work by Wei et al. is an intriguing numerical study that seeks to quantify how the number and distribution of inlet mouth islands impact the morphologic evolution of semi-enclosed coastal basins. The results of this investigation easily spur multiple investigations for future study, and it is evident that the work presented here could form the foundation of a fruitful line of research. As such, I support the publication of this paper, while acknowledging that some editing is needed to clean up grammar/syntax and add additional context to underscore the importance of the results. To this end, please see my comments below. Please note that, with regards to grammar, I have not flagged every instance of awkward wording and instead focus mostly on language that could impact comprehensibility.*

Reply: We wish to thank the reviewer for providing very constructive and detailed comments. We feel sorry for the confusion caused by some unclear wording in the manuscript. Following the

constructive comments, we have thoroughly modified our manuscript, rephrased many sentences, and made a clearer presentation of the model results.

*General Notes:*

*The authors should consider being more upfront about the fact that the model uses 'rocky' fixed-dimension islands in their model experiments. This detail is not apparent until section 2.3, although it is hinted at in the abstract ("geological constraints"). Stating this clearly in the abstract will allow other researchers to connect with this topic more specifically. Additionally, the Introduction switches between citations of geologically constrained and unconstrained systems, which is somewhat confusing when presented with model investigations that focus only on rocky mouth islands. I think this confusion could be eliminated by concentrating more on embayed and drowned coasts in the Introduction, with references to sandy/alluvial-type systems as points of comparison and to make more general observations about how mouth islands have broadly been observed to function. To be clear, I am not proposing a total teardown and rebuild of the Introduction, but more of a rearrangement and rephrasing to present observations in a way that could better support the specific investigations undertaken in this study. This may also make it easier to project future work in the Discussion, simultaneously giving the reader an early understanding about the limitations of this study while suggesting the importance of this work for subsequent investigation.*

Reply: Agreed. We have thoroughly revised and rephrased our abstract and introduction section to state 'rocky' islands much earlier. Further, we have rearranged our introduction section to concentre more on embayed and drowned coasts.

*Line Edits and Considerations:*

*Line 22 – 'Hindering role' in this sentence can probably remove to make the latter point in this sentence easier to understand. Recommend changing the dependent clause to "[…], while the presence of delta-side islands may increase relative sediment deposition in the basin."—or something similar, assuming I have captured the concept correctly.*

Reply: Yes, the reviewer is correct that the delta-side islands tend to increase relative sedimentation in the basin.

*Line 35 – The last part of this sentence should read either 'these types of coastal zones' or 'this type of coastal zone' to keep consistent with pluralization or lack thereof.*

Reply: Agreed and changed. We think 'this type of coastal zone' is better.

*Line 42 – Consider swapping 'changed landforms' for just 'landforms' to better convey that this is a two-way exchange of influence.*

Reply: Agreed and changed.

*Line 44 – The sentence describing the finding of Otvos (1981) is ambiguous. Without looking to the reference, my first thought reading this line is that it is possibly referring to sediment being liberated from an antecedent high to begin early barrier formation during sea-level rise. Please clarify.*

Reply: Yes, the reviewer is correct that Otvos (1981) explored the effects of sandbar drift on sandy island formation. Since this paper is more focused on rocky islands, we think it is better to remove this citation.

*Line 48 – The 'and' here is being used as a conjunction. Add a comma after 'systems.*

Reply: Agreed and modified.

*Line 49 – Consider splitting this sentence after 'tool'. 'Virtual' is also implied by numeral modeling, so not needed.*

Reply: Agreed and modified accordingly.

*Line 83 – The Massachusetts coast is heavily glacially-influenced, and bay islands are a mix of drowned uplands, glacial deposits, and alluvial features. To make the last part of the sentence more broadly accurate, could probably change to […], which are probably formed from the drowning of topographic highs during sea-level rise in the post-glacial period."*

Reply: Agreed and changed accordingly.

*Line 91 – When setting up the specific research questions, note that the number of islands is also being evaluated.*

Reply: Yes, the reviewer is correct that this paper has been designed to explore the effects of the number of islands.

*Line 143 – Merge with the previous paragraph.*

Reply: Agreed and changed.

*Line 152 – Suggest dropping 'Besides' from the beginning of the sentence. This line may also work better as the final sentence of the paragraph since it is a projection for future work, e.g. "In all cases, the same initial bathymetry is adopted so that the model results can be compared. Additionally, the islands in this initial study are non-erodible (rocky) and square (1 km x 1km). In the future, different sizes and shapes of islands will be investigated to determine how these parameters impact morphological outcomes."*

Reply: Agreed and modified accordingly.

*Line 165 – Need a conjunction to link thoughts about where currents occur, something like… "Morphological evolution first occurs in the mouth zone where tidal currents are strongest, as the well as the river input zone due to fluvial input."*

Reply: Agreed and modified accordingly.

*Line 173 – Add comma after 'therein'.*

Reply: Agreed and changed.

*Line 182 – Need an article before 'larger spatial scale'. Suggest 'a larger spatial scale'.*

Reply: Agreed and modified.

*Line 183 – Consider changing 'And more erosion occurs' to 'Also, more erosion occurs' or something similar.*

Reply: Agreed and changed.

*Line 184 – Slight phrasing corrections are needed here. Suggesting the following: "Meanwhile, in the upstream zone, small differences are observed between the four cases, indicating that hydrodynamic effects on this area are relatively limited."*

Reply: Agreed and modified accordingly.

*Figure 4: In the future, consider using a color palette that is more accessible to readers with color deficiencies. Again, not an issue here, but something to think about for the next study.*

Reply: Thanks for the good suggestion. We will pay more attention to this issue in the next study.

*Line 194 – I think "leading to more sediment suspended and transported, and forming a deeper inlet channel" could be rephrased more simply as just "leading to more suspended sediment transport and forming deeper inlet channels".*

Reply: Agreed and changed.

*Line 199 – Reading this paragraph reminds of how much this setup mimics the processes that occur around bridge pilings, which seems to be what the modeled rocky islands act like. Could be a thought worth mentioning in the discussion.*

Reply: We fully agree that the role of island chains at the marine scale looks somewhat similar to the role of bridge piles at the embayed scale. They both act as a converging and narrowing effect, thus resulting in higher water levels as well as stronger flows. We have mentioned this good idea in the discussion.

*Line 213 – Slight phrasing corrections are needed here. Suggesting the following: "As the morphological evolution continues, the channel gradually develops into the upper intertidal area and forms a complex channel network."*

Reply: Agreed and changed accordingly.

*Line 214 – Considering flipping the word order of 'scenarios of inlet island ("IL")' to 'inlet island scenarios'. This reordering would apply to similar phrasing throughout the paragraph.*

Reply: Agreed. We have revised similar phrasing throughout the manuscript.

*Line 219 – It is generally inadvisable to use contractions in a professional paper. Please change 'that's' to, in this case, 'This is'.*

Reply: Thanks for the good suggestion. We will pay more attention to this issue.

*Line 224 – Not sure what this is trying to say. Based on the previous sentence, this should say something about how morphology differs at CS3 with the longitudinal placement of the island, but as currently written there's just a broad statement about where channels and deposition generally develop. Please elaborate further.*

Reply: We feel sorry for the confusion caused by the unclear sentences. We have added more content to the revised manuscript to explain the different morphologies at CS3, as follows:

"In terms of ebb-delta area (CS3), the morphology is also significantly influenced by the longitudinal placement of the island. The inlet island scenario ("IL") develops several tidal channels in the ebb-delta area due to the diversion created by the inlet island (Figure 6a). In the BS and DS cases, the ebb-delta area has a similar morphology, with extensive deposition developing in the middle of the ebb delta. However, larger sedimentation occurs in the BS case, suggesting that it produces more significant sediment transport (Figure 6k, 6l)."

*Line 234 – "evolution a basin" should be "basin evolution".*

Reply: Agreed and modified.

*Line 248 – In the final sentence of this paragraph, I think the last line should read "[…] indicating that hydrodynamics gradually adapts to basin morphology and a relative equilibrium state." Please confirm if this is the correct meaning.*

Reply: Yes, the reviewer is correct that hydrodynamics gradually develops towards a relative equilibrium state after 300 years. This sentence has modified accordingly.

*Figure 8: This is my favorite figure in the paper!*

Reply: Thanks!

*Line 291 – "the ones gradually move the left side and become convex after 500 years" could be reworded to be more technically accurate. Perhaps something like "modeled hypsometric curves at 500 years become noticeably convex".*

Reply: Agreed and modified accordingly.

*Line 294 – Check the wording on this paragraph—the topic sentence, especially, does not really explain the result. If I understand correctly, there are really only two points here: [1] During basin morphological evolution, the area of tidal flats grows slightly slower under the delta-side scenario and slightly faster under the basin-side scenario. [2] After 500 years, the magnitude of shoals and flats developed under all cases are similar. These two thoughts can probably be appended to the previous paragraph.*

Reply: Yes, the reviewer is correct that these two points are really what we trying to make. We have rephrased this paragraph and appended it to the previous paragraph.

*Figure 9: Both the 100-yr and 500-yr curves being dashed is slightly confusing to look at. Consider either making one set solid lines, or alternatively, changing the color shading between the sets. I am thinking something along the lines of dark blue and light blue for BS, dark green and light green for DS, etc.*

Reply: Very good suggestion. We have modified the curve types and widths to make them easier to distinguish between cases. The new figure is shown below:

[Figure]

Figure 9: Hypsometry of the tidal basin for the simulations of different scenarios: (a) the scenarios of the different mouth island numbers; (b) the scenarios of different mouth island locations after 100 and 500 years respectively. The tidal amplitude in all cases is 1.2 m.

*Line 335 – Suggest changing "the tidal basin tends to be stable showed a gradually stable tidal prism" to "the tidal basin tends to be stable, as shown by an increasingly stable tidal prism."—or something similar.*

Reply: Agreed and modified accordingly.

*Line 405 – The second part of the last sentence in this paragraph does not really add anything to the discussion here. In light of the previous sentences in said paragraph, it reads almost contradictory, although I can see that is not the intent. Consider removing and/or elaborating.*

Reply: Agreed. We have removed the sentence in the revised manuscript.

*Section 4.2: This section could be better served in section 3, as it is effectively another result and could be easily discussed in the context of section 4.1. Consider rearranging—I think this would be fairly easy to implement and would improve the flow of the Discussion.*

Reply: Agreed. We have rearranged section 4.2 into the results section and have appropriately discussed it in the discussion section.

*Line 451 – In this first line of this section, remind the reader again that this is referring specifically to the basins in Massachusetts.*

Reply: Agreed and modified accordingly.

*Line 469 – I think it is possible these two paragraphs can be combined. The theme here is that the study does not fully capture the differences in the observed basins, which is probably a result of other geological constraints (particularly, initial bathymetry) that were not fully explored in this investigation. This could be used to set up a future sensitivity study to compare the relative magnitudes of morphological forcing from initial bathymetry and mouth island presence/placement.*

Reply: Agreed and combined accordingly.

*Line 475 – "while natural islands are often slowly eroded with time"—for the case of rocky islands specifically? Sandy mouth islands can be completely destroyed and reformed over decadal timescales. Might be worth adding some additional specificity.*

Reply: Yes, we fully agree with the reviewer that it is necessary to add some specificity for rocky islands. The erosion rate varies between different types of islands. Sandy islands can be completely destroyed and reformed over decadal timescales (Vousdoukas et al., 2020). Rocky islands can be eroded in a range of 0.01-0.1 myr-1, which largely depends on mechanical wave action and rock strength (Andriani and Walsh, 2007).

*Section 4.3: Overall, in this section, I was expecting more focus on future work, since the authors mention in section 2.3 that they have designs for more study. In general, I feel like this section could be more optimistic about the model results—this is a topic that is largely unexplored, and the fact that the basins which inspired the study have some differences that are not easily explained speaks to the richness of possible investigations in this research area.*

Reply: We highly appreciate this reviewer for the constructive and insightful comments. We feel all the comments substantially help to result in a much better manuscript, as well as benefit our next study. Aspects that are not well reproduced appear to relate to processes that have been omitted (e.g., initial bathymetry and sediment composition) and would merit further investigation.

*Conclusions: Depending on if suggestions for more future work discussion/speculation are considered in section 4.3, consider adding a corresponding line or two here.*

Reply: Agreed. We have added a corresponding line in this section.

**References**

Andriani, G. F. and Walsh, N.: Rocky coast geomorphology and erosional processes: a case study along the Murgia coastline South of Bari, Apulia—SE Italy, Geomorphology, 87, 224-238, https://doi.org/10.5281/10.1016/j.geomorph.2006.03.033, 2007.

Vousdoukas, M. I., Ranasinghe, R., Mentaschi, L., Plomaritis, T. A., Athanasiou, P., Luijendijk, A., and Feyen, L.: Sandy coastlines under threat of erosion, Nature Climate Change, 10, 260-263, https://doi.org/10.1038/s41558-020-0697-0, 2020.